



**The Dual-Edged Role of Vegetation in Evaluating Landslide**
**Susceptibility: Evidence from Watershed-Scale and Site-Specific**
**Analyses**
**Songtang He[a, c*], Zhenhong Shen[e], Jeffrey Neal[c], Zongji Yang[a], Jiangang Chen[a],**
**Daojie Wang[a], Yujing Yang[a,b], Peng Zhao[a,b], Xudong Hu[d], Yongming Lin[e],**
**Youtong Rong[c], Yanchen Zheng[f], Xiaoli Su[c], Yong Kong[g]**
[a]Key Laboratory of Mountain Hazards and Engineering Resilience, Institute of
Mountain Hazards and Environment, Chinese Academy of Sciences, 610213, Chengdu,
China
[b]University of Chinese Academy of Sciences, 101408, Beijing, China
[c]School of Geographical Sciences, University of Bristol, BS8 1SS, Bristol, UK
[d]College of Civil Engineering and Architecture, China Three Gorges University,
443002, Yichang, China
[e]College of JunCao Science and Ecology, Fujian Agriculture and Forestry University,
China, 350002, Fuzhou, China
[f]School of Civil, Aerospace, and Design Engineering, University of Bristol, BS8 1TR,
Bristol, UK
[g]Department of Civil and Environmental Engineering, The Hong Kong Polytechnic
University, Kowloon, Hong Kong SAR, China.
[*]Corresponding author. E-mail: hest@imde.ac.cn
Address: No.189, QunXianNan Street, TianFu New Area, Chengdu, Sichuan, China.



## Abstract

Vegetation is widely recognized for its beneficial role in landslide mitigation. However, shallow landslides frequently occur even in densely vegetated regions, suggesting that the influence of vegetation on gravity-driven erosion hazards remains incompletely understood. This study investigates the interactive effects of vegetation and key environmental factors—including rainfall, lithology, wind speed, and slope gradient—on landslide susceptibility in an area with substantial vegetation cover (≥65.5%). At the watershed scale, we employed structural equation modeling and geographic detectors to assess the primary drivers of landslide susceptibility under high vegetation conditions. At the point scale, we calculated the stability coefficient of a representative landslide, accounting for both vegetation self-weight and artificial waste sediment. Our findings reveal that the combination of vegetation, rainfall, and wind speed significantly increases landslide susceptibility, as evidenced by a 21.3% rise in high and very high susceptibility zones and a 42.7% reduction in low and very low susceptibility zones. Interactions among multiple factors exerted a stronger influence than individual factors, with the most pronounced interaction observed between slope gradient and rainfall (Geodetector q = 0.81), followed by rainfall and lithology (q = 0.79). Under saturated conditions, the stabilizing effect of root systems was outweighed by the self-weight of tree vegetation, leading to a marked decrease in slope stability compared to scenarios without additional loading. These results offer new insights into the complex role of vegetation in landslide control and highlight the importance of considering interactive environmental effects at multiple spatial scales.





**Keywords**: shallow landslide; high vegetation mountainous area; landslide
susceptibility; structural equation model; vegetation self-weight





## 1 Introduction

The prevalence of landslide hazards in regions with high vegetation cover suggests complex mechanisms in the relationship between vegetation and landslides (He et al., 2017; Xu et al., 2024), which extend beyond a linear association (Lan et al., 2020). Therefore, more research is needed to understand how slope stability and environmental factors interact under good vegetation cover conditions (Cui et al., 2024; Deng et al., 2022; Medina et al., 2021). Accurate hazard assessment and mitigation depend on understanding the interactions between vegetation, slope morphology, and landslide occurrence (Alvioli et al., 2024; Zhang et al., 2025)

Vegetation cover inhibits soil erosion, regulates soil moisture, and stabilizes slopes via root networks that anchor the soil (He et al., 2017), and ultimately improves slope morphology and soil structure to stabilize the slope. Therefore, higher vegetation cover is generally believed to help prevent landslide hazards (Rey et al., 2019). However, a recent work reposted that vegetation can reduce gully erosion but promote shallow landslides (Xu et al., 2024). Thus, vegetation can also increase landslide susceptibility by adding weight and altering soil properties (Qin et al., 2024). For example, trees represent a balance between root reinforcement and the destabilizing effect of tree weight, influenced by slope gradient (Schmaltz & Mergili, 2018). Additionally, wind forces can destabilize steep slopes (Bordoloi & Ng, 2020). On thin-soil slopes, root wedging may cause fractures, increasing susceptibility to external disturbances (Liu et al., 2020). Rainfall, a major trigger of landslides (Dhanai et al., 2022), saturates soil on slopes, reducing soil cohesion and shear resistance. When the shear strength is



insufficient to counteract the downslope forces generated by gravitational water
distribution, instability occurs, amplifying the destructive effects (Li et al., 2025). Areas
with high vegetation cover typically have favorable moisture and temperature
conditions. Although vegetation roots play an effective role in stabilizing soil,
vegetation can also alter soil properties by increasing soil moisture content or reducing
soil density, which affects matrix cohesion (Gonzalez-Ollauri & Mickovski, 2016;
Murgia et al., 2022; Vergani et al., 2017). Therefore, in areas with high vegetation cover,
the occurrence of landslide hazards is influenced by multiple spatial factors, such as
soil texture, vegetation, and rainfall. Investigating which factors primarily drive the
initiation (Xu et al., 2024), and how the combined effects of these factors influence the
occurrence and development of landslides, has become a key issue in landslide
forecasting and risk zoning. Consequently, further study of the relationship between
vegetation with environmental factors, and landslide is of significant importance.
Early studies used geoscience factor weights, logistic models, and Geographic
Information Systems (GIS) spatial analysis to assess the environmental factors that are
responsible for landslides (Regmi et al., 2010; Yilmaz, 2009). While these revealed
spatial distribution patterns, they focused on qualitative descriptions and analyses and
rarely addressed spatial correlations or interactions between factors. Few studies have
thoroughly explored the spatial correlation of factors contributing to landslides and
debris flows, resulting in a lack of quantitative representation of spatial correlations in
disaster risk. Research on vegetation types, height, and growth conditions (such as slope
and human disturbances) in relation to landslide risks remains limited. While,



significant progress has been made on the relationship between rainfall and landslides
(Ortiz-Giraldo et al., 2023), using models based on spatial autocorrelation (for example,
Moran's I and Geary's C indices) and clustering methods (Chen et al., 2024; Liu et al.,
2024; Pokharel et al., 2021; Schmaltz & Mergili, 2018; Wang et al., 2020), the
mechanisms of interaction between these factors remain unclear particularly the
modifying influence of vegetation on slope stability coefficients (Lan et al., 2020).
Overall, the dynamic processes through which vegetation and environmental factors
interact to shape landslide occurrences are still not well understood. To clarify how
rainfall, slope, lithology, and soil thickness influence landslides in well-vegetated
areas—and to explore vegetation's "double-edged sword" role in landslide mitigation—
this study examines the relationship between environmental factors and landslide
occurrence by integrating watershed-scale analysis with point-scale mechanistic
insights. In summary, focusing on the frequent occurrence of landslides and debris
flows in areas with high vegetation cover, this study selects a vegetation-rich region in
Southwest China as the study area. Through field investigations and remote sensing
image analysis, the study employs the Geodetector method and structural equation
modeling to analyze the influencing factors and mechanisms of landslide susceptibility
under a high-vegetation background at a macroscopic scale. Meanwhile, at the
microscopic level, it examines the role of vegetation as a mediator in coupling with
other factors (wind force, slope gradient, and load) in typical landslide occurrences
within the region. By integrating watershed scale and typical point landslide analyses
for mutual verification and interpretation, this study lays the foundation for elucidating



the dual role of vegetation in landslide prevention and mitigation.

## 2 Methods and materials

### 2.1 Study area

The study area is in Jinkouhe District, Sichuan Province, China (102°50′24″–
103°10′24″ E, 29°00′24″–29°00′46″ N). The Dadu River flows in a north-south "S"
shape through the 598 km² region, which is 99% mountainous, with complex geological
structures. Rock types include slate, clastic rocks, dolomite, limestone, and basalt.
Forest coverage is 65.55%, and the terrain slopes from southwest to northeast with
elevations ranging from 530 to 3,321 m, and a vertical drop of 2,700 m (**Fig. 1**).
Jinkouhe District has a subtropical monsoon climate with a warm and humid
atmosphere, abundant sunshine, and distinct seasons. The annual average temperature
is 15°C, with highs of 38°C and lows of -9°C. Precipitation averages 1252.6 mm
annually, concentrated in the summer, while average annual evaporation is 677.3 mm
with dry winters, spring droughts, frequent summer floods, and prolonged autumn
rainfall.



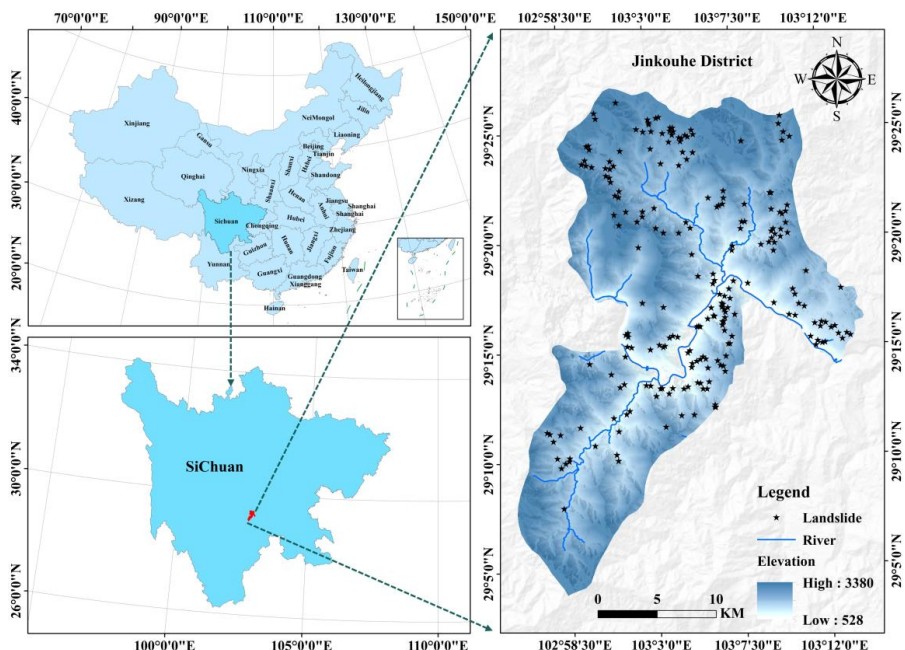

**Fig. 1. Location map of Jinkouhe District**

A major landslide occurred in the study area on June 4, 2023, near the living quarters of a Phosphate Mine (103°2′25.75″ E and 29°25′0.6″ N). According to the 2021 geological disaster risk assessment report for the Jinkouhe District, the collapse site is located within a high geological disaster susceptibility and medium-risk zone. The disaster was preceded by continuous rainfall.



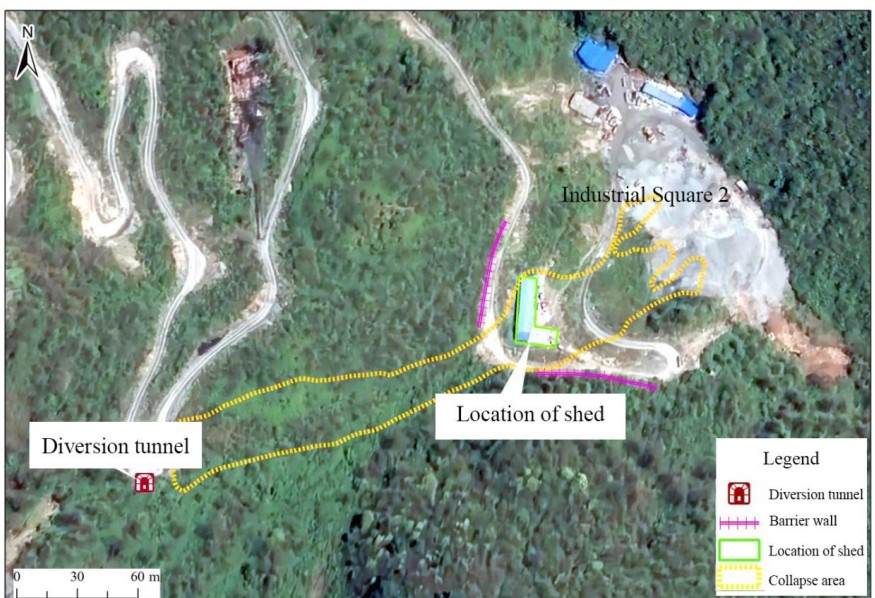

**Fig. 2. Schematic of Satellite Image Before Landslide Occurrence (basic picture from © Google Earth)**

In front of the collapsed area were the mine's living quarters. To the left, 70 m away, was the entrance to Tunnel No. 2, and 15 m behind it was a water diversion tunnel entrance (**Fig. 2**). During the construction of the diversion tunnel, 400 m³ of waste rock was deposited, adding approximately 800 tons of load, bringing the slope in the instability initiation zone to a critically stable state. The disaster was preceded by continuous rainfall. On May 31, 2023, moderate to heavy rainfall in the Jinkouhe area caused surface overland flow in a 0.1 km² catchment area at the top of the gully Vegetation increased water storage, and surface runoff infiltrated the Cambrian lower formation dolomitic limestone, generating significant groundwater. This groundwater flowed down along fractures and bedding planes but was obstructed by the sandstone and shale aquitard raising the groundwater level. Saturation of gravelly clay at the basal



interface softened the soil, reducing its shear strength and forming a slip surface,
triggering landslide. The shape of the landslide was elongated, resembling a "tongue."
The elevation ranged from 2,303 m at the rear to 2,183 m at the front, a height difference
of 120 m. The slope was 308 m long, and the collapse width ranged from 18 m to 41 m,
averaging 36 m. The maximum height of the rear scarp was 12 m. The collapse created
a residual body and deposition area, covering a total area of 10,186 m². The collapse
direction was 80°, with scouring in the center lateral deposits forming at the front edge
(**Fig. 3**).

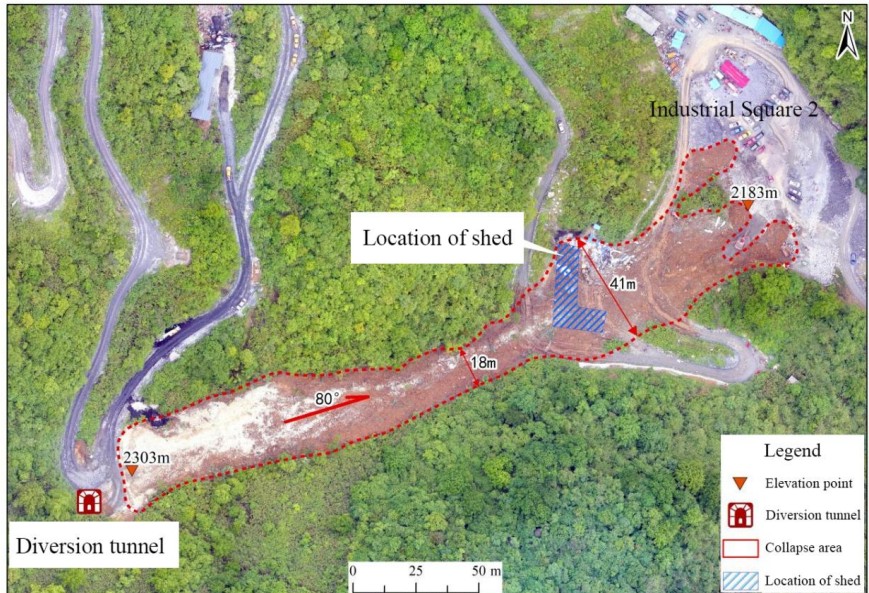


**Fig. 3. Schematic of the UAV image after the landslide**
**2.2 Data source and preprocessing**
The following data are required for landslide susceptibility analysis and spatial
correlation of triggering factors in the study area.



(1) **ASTER GDEM Digital Elevation Model** data with a resolution of 30 m were
obtained from a Geospatial Data Cloud (https://www.gscloud.cn/). Slope and aspect
factor data were extracted using surface analysis.
(2) **Road and river vector files** were obtained from Open Street Map
(https://www.openstreetmap.org/).
(3) **Lithology data** (Hengl et al., 2017) with a resolution of 250 m were acquired
from NASA's Open Data Center (https://search.earthdata.nasa.gov/).
(4) **Fault data** were obtained from the National Earthquake Science Data Center
(https://data.earthquake.cn/).
(6) **Maximum NDVI raster data** for China in 2021 with a resolution of 30 m and
2021 wind speed data with a resolution of 1 km (Xu, 2022a, 2022b) were obtained from
the Resource and Environment Science and Data Center of the Chinese Academy of
Sciences (https://www.resdc.cn/).
(7) **Monthly rainfall raster data** from 2018 to 2023 with a resolution of 1 km
(Peng, 2024), obtained from the National Tibetan Plateau Data Center
(https://data.tpdc.ac.cn/). To eliminate the random effects of precipitation, the multi-
year average annual precipitation was used as the influencing factor. Annual
precipitation was first calculated using map algebra, and then the multiyear average
annual precipitation was derived.
(8) **Landslide hazard point data** were obtained from the GeoCloud platform
(https://geocloud.cgs.gov.cn/). Additionally, 227 landslide points were identified
through the manual interpretation of remote sensing images.



All datasets were unified to the WGS_1984_UTM_Zone_47N coordinate system
and the pixel size was standardized to 30 m × 30 m. The processed results are shown in
Fig. S1.
**2.3 Landslide susceptibility evaluation framework**
**2.3.1 landslide susceptibility based on the analytic hierarchy process (AHP)**
Landslide susceptibility analysis was conducted by overlaying landslide sites with
elevation, slope, aspect, distances to faults, rivers, roads, lithology, rainfall, Normalized
Difference Vegetation Index (NDVI), and wind speed. All factors passed the
multicollinearity test with variance inflation factor (VIF) values below 10 (Arabameri
et al., 2019; Chen et al., 2018). Range normalization was applied to standardize all
indicators (He et al., 2024). A comprehensive multi-factor evaluation combined weight
calculations and GIS-based methods, including buffer, statistical, and overlay analysis.
Factors were classified and assigned weights, and validated using consistency tests
(Table S1). The overlay analysis produced a landslide susceptibility distribution map of
the study area (Ahmad et al., 2023; Asmare, 2023), with susceptibility categorized into
five levels: very low, low, medium, high, and very high, based on existing standards.
The landslide susceptibility index (SI) is calculated as:

$$SI = \sum w_i IF_I \tag{1}$$

where *SI* is the comprehensive geological hazard susceptibility index for the
evaluation unit; $w_i$ represents the weight of the influencing factor; and $IF_i$ represents the
value of the influencing factor.
The process involved altering the influencing factors to examine how





environmental variables (rainfall, vegetation, and wind speed) affect landslide
susceptibility. Common factors included elevation, slope, aspect, lithology, and
distances to faults, rivers, and roads. Vegetation, rainfall, and wind speed were added
successively, resulting in five scenarios.

Class I: Common factors.

Class II: Common factors + vegetation.

Class III: Common factors + rainfall.

Class IV: Common factors + vegetation + rainfall.

Class V: Common factors + vegetation + rainfall + wind speed.

The consistency ratio (*CI*) values for the five scenarios were 0.07, 0.09, 0.07, 0.08,

and 0.07, respectively, all of which were < 0.1, indicating that they passed the
consistency test (Table S2). Table S3 presents the eigenvectors and weights. The
formulas for calculating the Consistency Index and Consistency Ratio are as follows:

$$CI = \frac{\lambda_{max} - n}{n - 1} \qquad (2)$$


$$CR = \frac{CI}{RI} \qquad (3)$$

Where: RI is the random index (from the lookup table);$\lambda_{max}$ is the maximum
eigenvalue, and n is the number of factors.
**2.3.2 Rationality validation of susceptibility assessment results**

In this study, Receiver Operating Characteristic (ROC) curves and area under the

curve (AUC) values were used for validation. ROC curves provide a representation of
the specificity and sensitivity of an analytical method (Khosravi et al., 2019; Yilmaz,
2009). The AUC measures model accuracy, ranging from 0.5 to 1, with values closer to




1 indicating higher accuracy (Sezer et al., 2010).

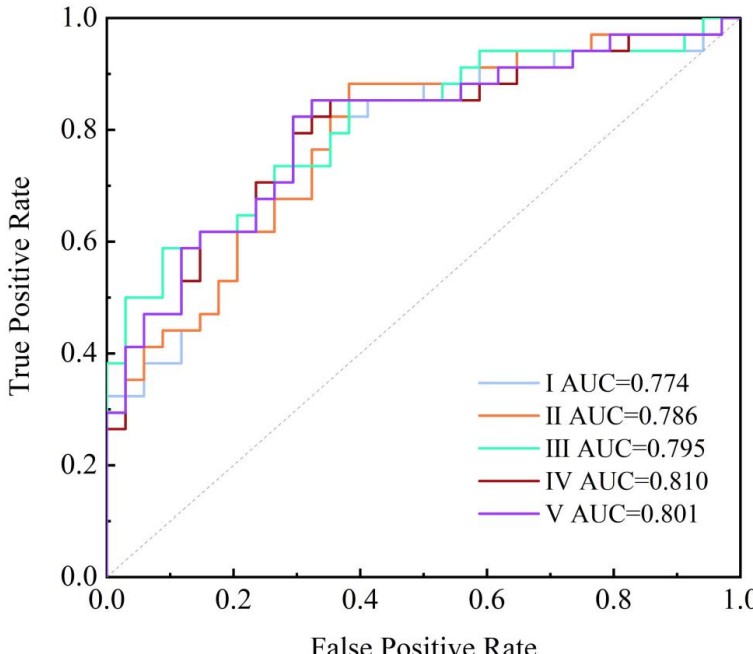


**Fig. 4. ROC curve of geological hazard susceptibility assessment results**

Based on the susceptibility distribution map, landslide point data obtained from

the Geological Cloud platform were selected, and an equal number of non-landslide
points were created to plot the ROC curve for landslide susceptibility. AUC values for
the five scenarios were 0.774, 0.786, 0.795, 0.81, and 0.801, respectively, all exceeding
0.5, indicating good accuracy in the landslide susceptibility evaluation (**Fig. 4**).
**2.4 Landslide driving mechanism analysis based on GeoDetector and structural**
**equation model**

The occurrence of landslides is associated with numerous factors. To identify the

dominant ones influencing landslide susceptibility, the GeoDetector method was
applied. This method examines the spatial differentiation of individual variables and



the interaction effects of two factors on the dependent variable, revealing their
explanatory power regarding landslide susceptibility. Furthermore, a structural equation
model (SEM), with its ability to analyze complex relationships among variables, was
employed to reveal how various factors interact to trigger landslides. (See **Fig. 5**).

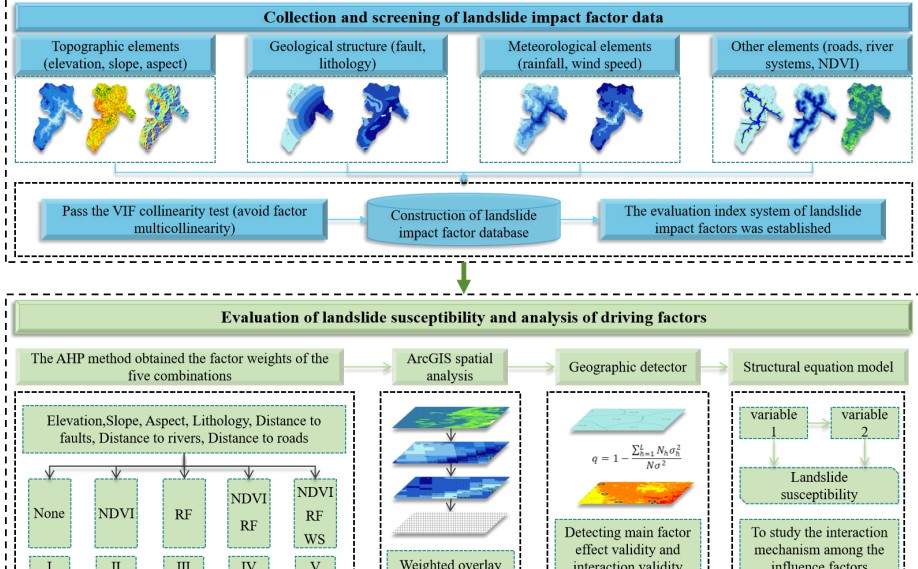


**Fig. 5. Technical Workflow Diagram**
**(1) GeoDetector**
Geo detection identifies spatially stratified variations and patterns of geographic
phenomena, addressing spatial dependence and heterogeneity caused by scale changes
that traditional statistical methods cannot resolve (Ng et al., 2021; Wang et al., 2010).
This study applied factor and differentiation to explore drivers of landslide
susceptibility in high vegetation areas. In single-factor detection, landslide
susceptibility was the dependent variable, and each evaluation factor was used as an
explanatory variable to identify key drivers influencing geological hazard risk (Lu et



al., 2024; Yang et al., 2024). Interaction detection evaluated whether the combined
effect of two factors increased or reduced explanatory power compared to their
individual effects, revealing interactions between variables. The influence of factors is
quantified by the *q*-value (0 to 1), with values closer to 1 indicating stronger spatial
differentiation explanatory power. The formula is:

$$q = 1 - \frac{\sum_{h=1}^{L} N_h \sigma_h^2}{N\sigma^2} \tag{4}$$

where *h* denotes the number of classifications or partitions for a specific indicator. *L*
represents the stratification of variable Y or factor X (i.e., classification or partitioning).
*N* is the total number of units in the study area. $N_h$ is the number of units in the *h-th*
stratum. $\sigma^2$ and $\sigma_h^2$ are the variances of landslide susceptibility in the entire study area
and the *h-th* stratum, respectively.
**(2) Structural equation model**
The SEM comprises two components: the measurement model, which defines
relationships between observed and latent variables, and the structural model, which
illustrates relationships among latent variables (Fan et al., 2016; Wang & Rhemtulla,
2021). Using GeoDetector results, SEM was used to analyze interactions among key
factors influencing landslide susceptibility. Based on literature and micro-mechanisms
(Chicas et al., 2024; Pourghasemi et al., 2018; Segoni et al., 2024), the following
hypotheses were proposed:
• Slope, lithology, distance to fault, distance to road, distance to river,
rainfall, and vegetation directly affect landslide occurrence
• Elevation and faults indirectly affect landslides by influencing vegetation





281 types and distribution

282 • Wind speed and river impact vegetation growth

283 • Rivers and elevation influence rainfall through the water vapour cycle

284 • Vegetation and wind speed indirectly influence landslide susceptibility by

285 influencing rainfall infiltration and local rainfall variations.

286 The SEM was constructed using RStudio software, and the path coefficients and

287 parameters were estimated using the maximum likelihood method.

288 **Measurement model:**

$$X = \Lambda_x \xi + \varepsilon_1 \tag{5}$$
$$Y = \Lambda_y \xi + \varepsilon_2 \tag{6}$$

289 **Structural model:**

$$\eta = B \times \eta + \Gamma \times \xi + \varepsilon_3 \tag{7}$$

290 where X/Y is the vector of exogenous/endogenous indicators. $\Lambda x$ / $\Lambda y$ is the factor

291 loading matrix of exogenous indicators for exogenous/endogenous latent variables. $\xi$

292 represents the exogenous latent variables. $\eta$ represents endogenous latent variables. $B$

293 represents the relationships among endogenous latent variables, $\Gamma$ indicates the effect

294 of exogenous latent variables on endogenous latent variables. $\varepsilon_1$、$\varepsilon_2$、$\varepsilon_3$ are the error

295 terms for exogenous indicators, endogenous indicators. The residual term represents

296 the unexplained portion of the endogenous latent variables within the equation. The

297 final model fit results are summarized in Table S4 (Hu & Bentler, 1999; Stone, 2021).

298 **2.5 Landslide stability calculation considering vegetation weight**

299 Based on the field survey, constructing the drainage tunnel added an additional

300 400 m³ of waste material, with a weight of approximately 800 tons, along with 500 trees,





each with an average weight of 60 kg. The collapsed mass was composed of gravel-
clay with a high moisture content, exhibiting a plastic state. The slopes natural unit
weight was $\gamma$ = 19.5 kN/m$^3$, and the saturated unit weight was $\gamma_w$ = 20 kN/m$^3$. The
natural shear strength was $c$ = 15.5 kPa, with an internal friction angle of 10.5°, and the
saturated shear strength was $c$ = 15.0 kPa, with the same friction angle. The slope
stability was calculated using the formula for tree-vegetated slopes from Lan et al.
(2022), accounting for the effects of artificial waste material and vegetation weight. The
formula is:

$$F_s = \frac{\sum(c_i l_i + G_i \cos \theta_i \tan \varphi_i + F_i)}{\sum(G_i + G_i') \sin \theta_I} \qquad (8)$$

where: $F_i$ represents the anchoring force of the vegetation's vertical roots; $G_i'$ denotes
the downward gravitational force exerted by the vegetation; $G_i$ refers to the vertical
gravitational force acting on the soil mass; i represents the length of the sliding arc, $\varphi_i$
is the internal friction angle, and $\theta_i$ is the angle between the i-th sliding block and the
vertical direction. The stress analysis of the slope, incorporating the weight of the
vegetation, is illustrated in Fig. 6.

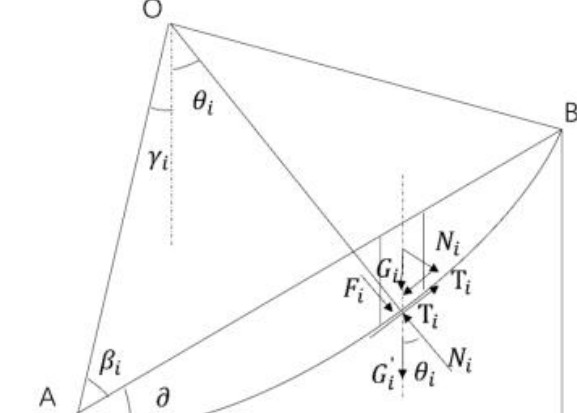




**Fig. 6. Slope Stress Analysis Diagram**

**3 Results**
**3.1 Landslide susceptibility mapping and distribution characteristics**
Fig. 7 and Table 1 present the landslide susceptibility results considering
environmental factors (rainfall, vegetation, and wind speed) along with common factors
(elevation, slope, aspect, lithology, distance to fault, river, and road). Most of the study
area shows moderate landslide susceptibility, with 84 landslides covering 53.77% of
the total area. Low and very low susceptibility zones contained 30 landslides covering
16% of the area. High and very high susceptibility zones, occupying 30.23% of the
study area, experienced 113 landslides.
Moderate susceptibility zones are widespread across the northern, western, and
southwestern regions. High and very high susceptibility zones, though smaller in
coverage, are primarily located in the central-eastern region. High susceptibility zones
display a "cross-shaped" spatial distribution, while very high susceptibility zones are
scattered within the high susceptibility areas.

**Table 1. Distribution of landslide susceptibility zones in Jinkouhe District**

| Susceptibility Zone | Number of Landslides (count) | Zoning Area (km²) | Proportion of landslide points in the total (%) | Proportion of zone to total area (%) | Landslide Point Density (points/km²) |
|---|---|---|---|---|---|
| Very low | 1 | 7.33 | 0.44 | 1.21 | 0.14 |
| Low | 29 | 89.43 | 12.78 | 14.79 | 0.32 |
| Mid | 84 | 325.06 | 37.00 | 53.77 | 0.26 |
| High | 105 | 167.26 | 46.26 | 27.67 | 0.63 |
| Very high | 8 | 15.49 | 3.52 | 2.56 | 0.52 |





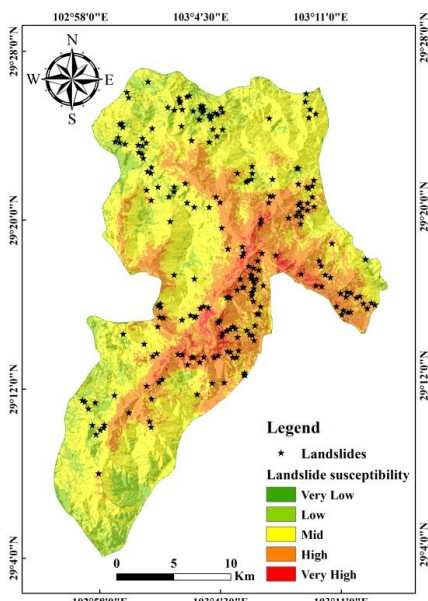


**Fig. 7.Landslide susceptibility assessment map**
**3.2 Relationship between H/L and area of a typical landslide**
Landslide parameters for each segment, including height, travel distance, and
landslide area (Table S5), along with overall landslide parameters (**Table 2**), were
measured. The total landslide height (H) is 120 m, with an average collapse width of
36 m. Travel distances increase sequentially across the three segments, with the
deposition zone having the longest at 110.9 m. The total landslide travel distance (L),
calculated as the sum of all segments, is 283.3 m.

**Table 2. Main morphological parameters of landslide**

| Data types | Number |
|---|---|
| Total landslide area (m²) | $1 \times 10^4$ |
| Total landslide volume (m³) | $3.1 \times 10^4$ |
| Total landslide length (m) | 308 |
| Landslide starting point elevation (m) | 2303 |
| Landslide endpoint elevation (m) | 2183 |
| Landslide height (m) | 120 |
| Landslide movement distance (m) | 283.3 |
| H/L | 0.42 |



As travel distance increases, the landslide area expands rapidly from the initial
instability zone (2147 m²), scraping away surrounding soil, and eventually forming a
deposition area of 5234 m². The final total landslide area was10,000 m².
Using the landslide height (H) and maximum travel distance (L), H/L was
calculated to be 0.42, which was less than 0.60, classifying it as a general or medium-
sized landslide. Although its activity intensity and destructive potential may be lower
than those of high-speed, long-runout landslides, its potential hazard level remains
moderate to relatively high (Text S1).
**3.3 Evaluation results of slope stability considering artificial waste sediment and**
**vegetation self-weight**
**Table 3** shows stability results under four conditions: natural and saturated states
without and with the loading of artificial waste sediment and vegetation self-weight.
Results indicate that the slope remained stable under natural and saturated conditions
without loading, with greater stability in the natural state. Under natural conditions with
waste sediment loading, the slope was stable. However, under saturated conditions,
with both waste and vegetation self-weight loadings, the slope became unstable and
failed. Comparing results revealed that the self-weight of materials in the natural state
was even greater than those in the saturated state.

**Table 3. Stability results under different conditions**

| Geotechnical condition | Without waste material and vegetation | | With waste material and vegetation | |
|---|---|---|---|---|
| | Natural Condition | Water-saturated | Natural Condition | Water-saturated |
| Stability Factor (Fs) | 1.21 | 1.13 | 1.02 | 0.89 |





## 4 Discussion

### 4.1 Analysis of landslide driving forces and mechanisms

Based on the distribution of landslide occurrences within different thresholds of the influencing factors (Text S2), landslides in areas with good vegetation conditions are controlled by multiple factors. **Table 4** shows the explanatory power of each factor for landslide susceptibility, ranked as follows: rainfall > elevation > distance to fault > distance to river > wind speed > lithology > slope > distance to road > NDVI.

**Table 4. Explanatory power of each factor on landslide susceptibility**

|  | X1 | X2 | X3 | X4 | X5 | X6 | X7 | X8 | X9 | X10 |
|---|---|---|---|---|---|---|---|---|---|---|
| q statistic | 0.55 | 0.28 | 0.06 | 0.40 | 0.49 | 0.28 | 0.46 | 0.27 | 0.63 | 0.44 |
| rank | 2 | 7 | 10 | 6 | 3 | 8 | 4 | 9 | 1 | 5 |

Note: X1–X10 represent elevation, slope, aspect, lithology, distance to fault, distance to road, distance to river, NDVI, rainfall, and wind speed, respectively.

Further analysis using interaction detection assessed the explanatory power of interactions between factors for landslide susceptibility (**Fig. 8**). Among 45 pairs of interacting factors, 40 exhibited bifactor enhancement, and five pairs showed nonlinear enhancement, with no independence or weakening observed. Bifactor interactions had a significantly stronger influence on landslide susceptibility than single factors. The most impactful interaction was between slope and rainfall (q=0.81), followed by rainfall and lithology (q=0.79). Rainfall interactions with other factors exceeded 70%, and fault distance interactions surpassed 60%. Wind speed interactions with elevation, slope, rainfall, lithology, distance to road, and NDVI exhibited bifactor enhancement all above 50%. Similarly, NDVI interactions with elevation, rainfall, lithology, distance to fault, road, and river, and wind speed also showed bifactor enhancement, with influence levels exceeding 50%. From the results of the factor interactions, the



synergistic effects of rainfall, fault distance, lithology, NDVI, and wind speed emerged
as the dominant interaction modes driving landslide susceptibility.

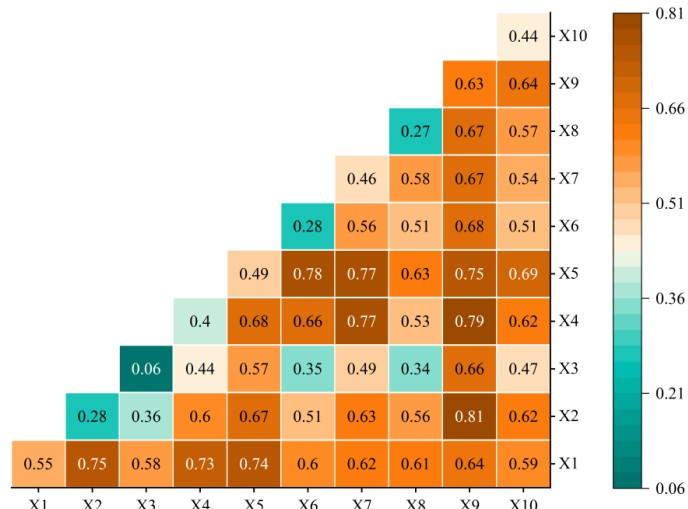


**Fig. 8. Explanatory Power of Interactions Between Factors on Landslide Susceptibility.**
Note:X1–X10 represent elevation, slope, aspect, lithology, distance to fault, distance to road,
distance to river, NDVI, rainfall, and wind speed, respectively.
To further clarify how various factors interact and contribute to landslide
occurrence, an SEM was constructed (Table S4) using the key factors identified by the
GeoDetector (excluding aspect). The SEM explained 92% of the landslide
susceptibility variance, with all factors showing significant correlations (p < 0.01) (**Fig.**
**9**, **Table 5**). The influence of each showed varying direct and indirect effects. Slope had
the greatest impact, with a total effect coefficient of 0.33, attributed to direct effects
(coefficient: 0.33). Steeper slopes increased the likelihood of landslides. The total effect
coefficient of distance to fault (0.30) was second only to slope, primarily reflected as a
direct effect (0.27), with a minimal indirect effect coefficient of 0.03. This is because
faults occur where the structural stability of the slopes is poor. Fault zones can cause





localized stress and weaken the structural integrity of slopes. This is particularly true in
the study area, which lies at the intersection of the Longquanshan Fault Zone and the
Ebian-Mabian Seismic Belt located in the central segment of China's north–south
seismic belt. Proximity to faults increases the susceptibility of rock and soil structures
to fault ruptures, thereby raising landslide susceptibility. Faults also allow water
infiltration, altering soil moisture and groundwater levels, and thereby influencing
landslide-triggering conditions. For instance, increased rainfall may elevate pore water
pressure in the soil and release through fault zone weak points, further heightening
landslide susceptibility.

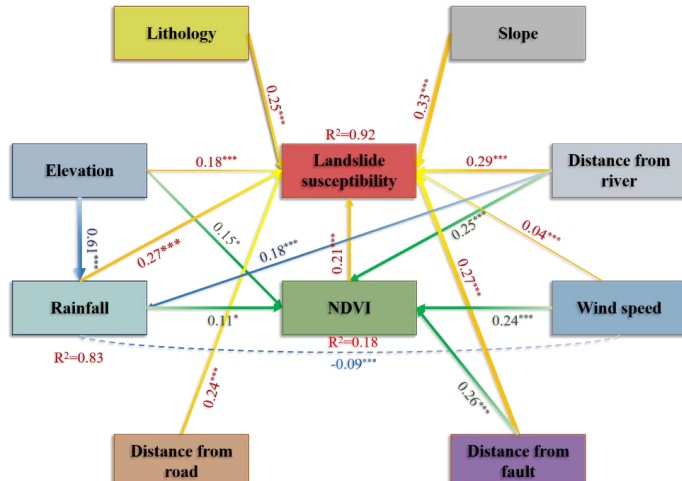


**Fig. 9. SEM of landslide susceptibility.**
Note: Rectangles = observed variables; unidirectional arrows = relationship between two variables.
The variable at the arrowhead is influenced by the variable at the arrow's base. The numbers near
the yellow single arrows represent the total effect coefficients of factors on landslide susceptibility.
Arrows in other colors indicate the standardized normalized influence coefficients between factors.
Solid arrows = positive = relationships; $R^2$ = proportion of variance explained. ***, **, and * denote
significance levels of 0.01, 0.05, and 0.1, respectively.
The total effect coefficient of the distance to the river was 0.29, primarily reflected



as a direct effect (coefficient: 0.2). The indirect effect (coefficient: 0.09) mainly
influences landslides indirectly through NDVI and rainfall. The total effect coefficient
of rainfall on landslides is 0.27, with a direct effect coefficient of 0.24. This is primarily
because rainfall increases soil moisture content and bulk density while reducing soil
shear strength. As soil moisture continues to accumulate, transient additional water
loads form, further weakening slope stability. Rainfall also generates surface runoff,
eroding slopes, damaging the slope structure, and thus increasing landslide
susceptibility. Its indirect effect (coefficient: 0.03), comes from groundwater recharge
and altered water flow paths modifying slope morphology over time and heightening
landslide risk.

**Table 5. Influence of various factors on landslide susceptibility**

| Factor | Total effect | Direct effect | Indirect effect |
|---|---|---|---|
| Elevation | 0.18 | 0 | 0.18 |
| Slope | 0.33 | 0.33 | 0 |
| Lithology | 0.25 | 0.25 | 0 |
| Distance from fault | 0.30 | 0.27 | 0.03 |
| Distance from road | 0.24 | 0.24 | 0 |
| Distance from river | 0.29 | 0.2 | 0.09 |
| NDVI | 0.21 | 0 | 0.21 |
| Rainfall | 0.27 | 0.24 | 0.03 |
| Wind speed | 0.04 | 0 | 0.04 |

In terms of direct effects, lithology has a total effect coefficient of 0.25 on
landslides, entirely direct. Lithology provides the material basis for landslides as soil
layers formed by different lithologies vary in shear strength and permeability. In the
study area, soft rocks, such as shale and clastic rocks, which became loose and highly
weathered after absorbing water, exhibited strong deformation and a high likelihood of
causing landslides. Roads had a total effect coefficient of 0.24, which was also entirely



direct. This is because road construction often involves slope excavation and vegetation
destruction, directly altering the stability of the natural terrain. Poorly designed road
drainage can also cause water to accumulate, affecting slope stability over time.

Regarding indirect effects, wind speed has an influence coefficient of only 0.04.

Generally, wind alone cannot directly destabilize a slope, but affects stability through
drag forces. Strong winds can strip soil cover, accelerating moisture evaporation,
leading to surface cracks and reducing structural stability. Higher elevations (effect:
0.18) imply more rainfall and weathering, but are not direct landside triggers. NDVI
had a total effect coefficient of 0.21 indirectly enhancing slope stability. through
vegetation roots which improve soil cohesion and shear strength. However, root
systems can cause deformation, increasing water infiltration and reducing slope stability.
This effect can be particularly pronounced in areas with dense, well-developed root
systems. Tall trees on steep slopes can reduce stability due to weight and wind drag.
Vegetation also indirectly regulates soil moisture through transpiration, reducing pore
water pressure, which affects slope stability.

In summary, vegetation mediates multiple factors, amplifying their combined

effects on landslide susceptibility.  Interactions between vegetation and other factors
are crucial in assessing landslide risks.
**4.2 Comparison of landslide susceptibility results under different influence factor**
**combinations**

It has been demonstrated that vegetation, wind, and rainfall have varying direct or

indirect impacts on landslide susceptibility. These impacts are based on the statistical




relationship between thresholds of influencing factors and the number of landslides, as
well as the contributions of various factors to landslide susceptibility through direct or
indirect effects analyzed using GeoDetector and SEM. The spatial distribution and
statistical data of landslide susceptibility under the influence of various environmental
factors (rainfall, vegetation, and wind) are depicted in **Fig. 10** and **Table 6**: taking into
account public factors alone (Category I), public factors + vegetation (Category II),
public factors + rainfall (Category III), public factors + vegetation + rainfall (Category
IV), and public factors + vegetation + rainfall + wind (Category V).

**Table 6. Distribution of landslide susceptibility zones for the five scenarios**

|  | I | | II | | III | | IV | | V | |
|---|---|---|---|---|---|---|---|---|---|---|
|  | Area | Quantity | Area | Quantity | Area | Quantity | Area | Quantity | Area | Quantity |
| Very low | 15.63 | 7 | 13.68 | 2 | 7.74 | 1 | 8.79 | 1 | 7.33 | 1 |
| Low | 150.19 | 43 | 155.30 | 50 | 100.09 | 24 | 123.91 | 39 | 89.43 | 29 |
| Mid | 284.81 | 87 | 291.75 | 87 | 313.12 | 97 | 303.06 | 80 | 325.06 | 84 |
| High | 128.85 | 78 | 133.11 | 83 | 150.40 | 85 | 152.51 | 97 | 167.26 | 105 |
| Very high | 25.08 | 12 | 10.72 | 5 | 33.21 | 20 | 16.30 | 10 | 15.49 | 8 |
| Total | 604.56 | 227 | 604.56 | 227 | 604.56 | 227 | 604.56 | 227 | 604.56 | 227 |

The results indicate that using Category I as the baseline, adding vegetation
independently (Category II) decreases very low and very high susceptibility zones by
14.37 km² and 1.95 km², respectively, with fewer landslides. Other zones saw slight
increases, with the low susceptibility zone increasing by 15.11 km² and seven additional
landslides, suggesting that vegetation inhibits landslides.
When rainfall is added independently (Category III), overall susceptibility
increases significantly. Very low and low susceptibility zones decrease by a total of



57.99 km², and the medium susceptibility zones increase by 28.31 km² with 10
additional landslides. The high and very high susceptibility zones increase by 29.69
km² with 15 additional landslides. This indicates that vegetation has a positive
inhibitory effect on landslide occurrences in favorable eco-geological conditions (such
as terrain, slope, and altitude). However, in high or very high susceptibility zones with
harsh conditions, rainfall amplifies vegetation's contribution to landslide susceptibility.

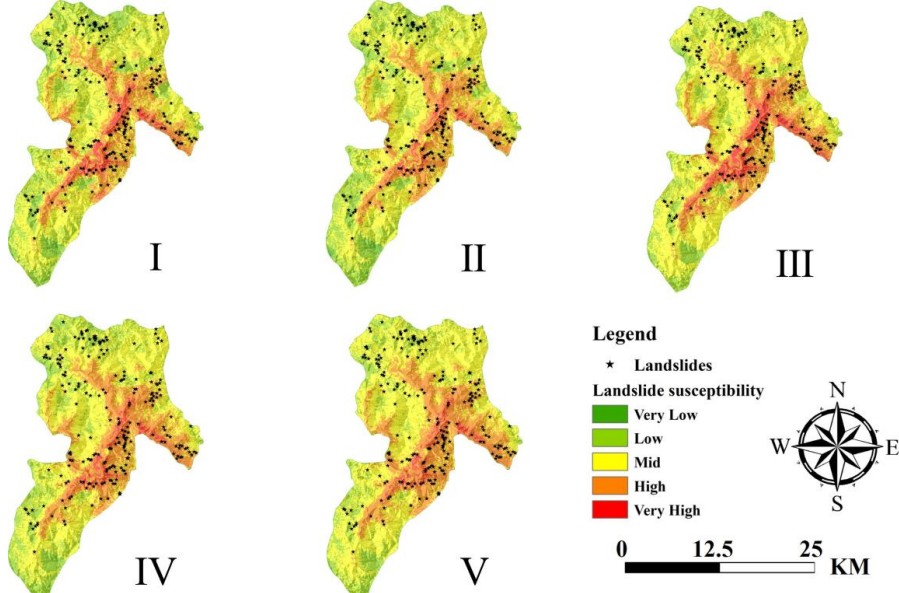


**Fig. 10. Landslide Susceptibility Distribution Map for Five Scenarios**

When rainfall and vegetation factors are considered together (Category IV), the

very low and low susceptibility zones decrease by 33.12 km², while the medium
susceptibility zones increase by 18.25 km², with ten fewer landslides. The high
susceptibility zone expands by 23.66 km², with 19 additional landslides. Although the
very high susceptibility zone shrinks in area, disaster point density increases to 0.61 per
km².Adding the wind speed factor (Category V) causes the very low and low



susceptibility zones to decrease significantly by 69.06 km²—almost double the
reduction observed in Category IV. The medium and high susceptibility zones expand
significantly by 40.25 km² and 38.4 km², respectively. Together, the high and very high
susceptibility zones experienced 23 more landslides.
Four patterns emerge: (1) In areas with low NDVI and rainfall, landslide
susceptibility shifts from very low to low; (2) In areas with high NDVI and slightly
lower rainfall, susceptibility changes from low to medium; (3) In areas with low NDVI
and high rainfall, landslide susceptibility shows little change; (4) In areas with high
NDVI and rainfall, susceptibility shifts from medium to high.
In summary, vegetation's role in landslide occurrence is influenced by external
environmental conditions. This aligns landslide stability models that consider the self-
weight of vegetation. It is essential to recognize the double-edged nature of vegetation
in landslide prevention and control.
**4.3 Mechanisms of landslides in areas with high vegetation coverage**
The mechanisms of landslide occurrence in areas with high vegetation coverage
represent an important yet underexplored topic. Good vegetation coverage can create a
false sense of safety and stability. First, Dense vegetation can obscure surface changes,
making it difficult to detect geological or erosion, leading to an underestimation of risks.
Secondly, while vegetation coverage can slow down rainwater flow and reduce its
impact during initial stages or under conditions of low rainfall over a short period, the
dense root networks increase soil stability, thereby enhancing the soil's resistance to
landslide initiation. However, areas with good vegetation coverage are often regions




with favorable water and thermal conditions. Vegetation's absorption of part of the
rainfall also increases soil saturation, further exacerbating the risks of landslides and
debris flows. Under persistent warm and humid conditions, geological and surface
changes in high vegetation areas are often gradual. Signs of disasters may accumulate
over long timescales, making these changes easy to overlook. Moreover, factors such
as seasonal rainfall or snowmelt can increase disaster risks during specific periods,
while other times may seem relatively safe. This seasonal variability adds uncertainty
to disaster occurrences, further enhancing their concealed nature.

Furthermore, terrain complexity is a key factor contributing to the concealed

nature of landslides in areas with high vegetation. These regions often feature rugged
terrain with significant elevation changes, which may hide numerous geological
structural issues. Such structures can lead to landslides under certain conditions, but
their prediction is challenging due to the terrain's complexity. For instance, localized
soil erosion or slope collapses might occur in these terrains, but the vegetation cover
can make such signs difficult to observe.

Where terrain and rainfall factors combine, vegetation can amplify disaster effects.

For instance, strong winds can destabilize vegetated slopes, increasing the risk of soil
erosion and landslides. Thus, the concealed mechanisms of landslides result from the
combined effects of geological, meteorological, and topographical factors.
Comprehensive analysis of these factors is essential to understand hidden risks.
Strengthening early warning systems and implementing preventive measures are
critical to reducing potential casualties and property losses.





**4.4 Comparison with previous studies and scope for future research**

Cui et al. (2024) analyzed the characteristics and causes of a similar landslide in

this area using Massflow V2.8 simulations. They identified rainfall and human
activities as key triggers, but insufficiently addressed interactions between soil,
moisture, and external forces (such as natural wind and human mining activities) under
high vegetation conditions. This limited simulation accuracy.

The current study uses macroscopic susceptibility mapping to elucidate the

interaction mechanisms among landslide susceptibility factors, with vegetation as the
core medium. Microscopic stress analysis of slope stability was used to calculate slope
stability while accounting for vegetation self-weight. The findings provide an important
reference for studying landslide movement characteristics and developing disaster
prevention and mitigation strategies in high vegetation mountainous areas.

Although this study analyzes landslides at the watershed scale and specific points.

further research is needed at regional, national and global scales to understand the
factors influencing both the positive and negative roles of vegetation in landslide
prevention. In particular, the specific causes, damage patterns, and impacts of the
negative effects remain unclear. On the basis of clarifying the interactions between
vegetation and environmental factors (such as rainfall, slope gradient, lithology, and
soil thickness), efforts should be made to construct regional or global zoning maps of
dominant factors for mountain hazards, considering vegetation and its synergistic
factors.

Additionally, while each of the conventional approaches to landslide susceptibility





assessment—such as the information value method, deterministic coefficient method,
and analytic hierarchy process (AHP)—have advantages, they also have drawbacks,
including high subjectivity, limited explanatory power, and limited applicability. It is
challenging to fulfill the present standards for high-precision and high-efficiency
landslide risk assessments because of these limitations. Machine learning technologies,
have been widely applied in landslide susceptibility assessments. LightGBM and
XGBoost have shown excellent performance in susceptibility assessments, but lack
interpretability, limiting their credibility.  Advances in interpretable machine learning
could integrate agent-based models to simulate spatial interactions of factors.
Combining these with tools such as GeoDetector and SEM would provide data and
form the foundation for developing effective landslide prevention strategies.

**5 Conclusion**

This study integrates regional landslide susceptibility mapping with stability

analysis of typical landslides, revealing spatial variation patterns and mechanisms of
landslide susceptibility in high vegetation areas. It clarifies the positive and negative
roles of vegetation in landslide prevention and control. While offering insights into the
role of vegetation in ecological disaster reduction, the landslide mapping in this study
relies on traditional methods. Although these methods have little impact on the overall
distribution patterns of landslide susceptibility, they exhibit limited precision to some
extent, particularly in small watersheds, where the evaluation results are coarse. With
the continuous advancement of machine learning techniques, leveraging these methods'
interpretability and strong learning capabilities can further improve the accuracy of



landslide susceptibility mapping in future studies. This study gives limited
consideration to factors such as earthquakes, forest height, and root distribution. By
incorporating more extensive datasets in future research, the changes in landslide
susceptibility and its driving forces can be better understood. In summary, this study
examines the role of vegetation in landslides from both macro and micro perspectives,
providing theoretical support for further landslide risk assessments.

### 581 Code and data availability

Data will be made available on request. Please contact the first author regarding the
availability of code and data used in this work.

### 584 Author contributions

**Songtang He:** writing – original draft, visualization, validation, data curation.
**Zhenhong Shen and Yuqing Yang:** software, methodology, formal analysis. **Jeffrey**
**Neal, Xudong Hu, and Yongming Lin:** writing – review & editing. **Zongji Yang,**
**Jiangang Chen, Daojie Wang:** investigation, supervision, resources. **Youtong Rong,**
**Yanchen Zheng, Xiaoli Su, and Yong Kong:** review & editing, data check, validation.

### 590 Competing interests

The contact author has declared that none of the authors has any competing interests.

### 592 Acknowledgements

This work was financially supported by the National Key R&D Program of China
(2024YFC3012702; 2024YFF1307801; 2024YFF1307800), the Youth Innovation
Promotion Association of the Chinese Academy of Sciences (Grant No. 2023389), the
National Natural Science Foundation of China (Grant No. 42201094; Grant No.



42371014), the Science and Technology Research Program of the Institute of Mountain
Hazards and Environment, Chinese Academy of Sciences (Grant No. IMHE-CXTD-
01), China Scholarship Council (CSC) for Academic Visiting (202404910203).
**Declaration of interest Statement**
The authors declare that they have no known competing financial interests or personal
relationships that could have appeared to influence the work reported in this paper.

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
