# Peer review of "The Dual-Edged Role of Vegetation in Evaluating Landslide 1 Susceptibility: Evidence from Watershed-Scale and Site-Specific 2 Analyses 3 4 Songtang Hea, c\*, Zhenhong Shene, Jeffrey Neale, Zongji Yanga, Jiangang Chena, 5 Daojie Wanga, Yujing Yanga,b, Peng Zhaoa,b"

_EGUsphere, 2025_

## Community Comment (CC1)

This work explores the "double-edged sword" role of vegetation in the landslide susceptibility, conducting multi-scale research at both the watershed scale and typical points, and is of certain innovation and academic value. Thus, to some extent, it is very meaningful. But some of content should be improved, the suggestions are show below. So, I give it moderate revision.

**Introduction**

The introduction provides sufficient background information regarding landslide hazards to introduce readers to the study. The gap in the literature, that more research is needed in regions with high vegetation, and the limitations of prior studies has been clearly explained. The motivation for the study and its objectives are stated clearly. But, also need some revisions just as follows:

1 The following studies cited are older than 10-15 years: Regmi et al., 2010; Yilmaz, 2009, Fell et al., 2008; Hürlimann et al., 2008; Sezer et al., 2010; Hu & Bentler, 1999; Goren et al., 2010; Manzella et al., 2008. It is preferable to cite recent articles in a manuscript, and only in exceptional circumstances should references going back more than 10-15 years be cited. It is preferable to cite recent articles because older references may be irrelevant given more recent advancements in the field of study. Exceptions to this rule should be reserved for seminal works directly relevant to the topic of research. Citing recent articles also helps journal editors see that there is a potential audience for your topic of research;

2 There is some repetition in the introduction, which can be frustrating for your readers;

3 The last paragraph is so long, please split into two part.

4 Line 67, "reposted" should be "reported".

**Methods and materials**

The methods present all the necessary information to be reproduced by other researchers, and the reasons for choosing specific methods have been included, where relevant. The methods are presented in a logical order. The procedure for statistical analyses of the data collected has been outlined under Methods. But followings should be paid more attention,

1 Figures 1,2 and 4 were referenced in the text **after** the appearance of the figure. Please amend;

2 Clarify why the specific study area (Jinkouhe District) was chosen—how do its

characteristics contribute to the relevance of this research?

3 Provide more detailed descriptions of the modeling processes, especially SEM, including assumptions made during factor selection.

4 please redraw the Fig. 6.

**Results**

The order of the results corresponds to that of the methods. The tables and figures present the data clearly and are referenced in the manuscript. Table and figure details corresponded with those in the text.

1 Lines 330-332, It would be useful to give the areas (north, southwest, etc.);

2 Line 335, please left a space between Fig.7.landslide…………

3 Lines 352-353, the title 3.3 can be revised as "Slope stability calculation considering artificial waste sediment and vegetation self-weight".

**Discussions**

This section provides a thorough discussion of the results-based part and the differences and improvements compared to previous studies, and also offers an outlook on future work.

However, it is necessary to more clearly point out the unique aspects of this research (for example, "integrating macroscopic susceptibility with microscopic mechanics"). The outlook for future research can be more specific, for example: how to utilize interpretable machine learning and multi-source data fusion, rather than just making general statements. More specific as below:

1 Line 364, the title should be changed to "Analysis of landslide driving factors and their interaction pathways". This part, the authors mainly emphasize the factors and the interactions.

2 Line 453, the title should be precise. This part mainly compared the landslide susceptibility under different factors combination, so maybe this title will be more suitable: "differences and explanations of landslide susceptibility results under different factor combinations".

---

## Author Comment (AC1)

**Response letter**

**Dear Editors and Reviewers:**

**Re: egusphere-2025-3004**

We sincerely thank you and reviewers for providing us with such a valuable revision opportunity. Thus, we can further improve and present our studies. The comments from you and the reviewers were highly insightful and enabled us to greatly improve the quality of our manuscript. We have carefully reviewed the feedback and made corrections that we hope will be met with approval. Revised portions are marked on the revised manuscript. Please note that these resulting revisions did not change the paper's findings.

In the response letter to editor and reviewers, we firstly summarized the major changes in a cover letter to editors, and we then itemized response to editors and reviewers, **in which the blue font indicates the response to each comment and the black font presents the revision from the revised manuscript.**

We hope that the revisions in the revised manuscript and the responses to the comments will suffice to allow our manuscript to be suitable for publication in ***Natural Hazards and Earth System Sciences***.

**Sincerely regards,**

**Songtang He (hest@imde.ac.cn)**

**Institute of Mountain Hazards and Environment, Chinese Academy of Science**

**Response to Reviewer #1**

**[Comment 1]** The following studies cited are older than 10-15 years: Regmi et al., 2010; Yilmaz, 2009, Fell et al., 2008; Hürlimann et al., 2008; Sezer et al., 2010; Hu & Bentler, 1999; Goren et al., 2010; Manzella et al., 2008. It is preferable to cite recent articles in a manuscript, and only in exceptional circumstances should references going back more than 10-15 years be cited. It is preferable to cite recent articles because older references may be irrelevant given more recent advancements in the field of study. Exceptions to this rule should be reserved for seminal works directly relevant to the topic of research. Citing recent articles also helps journal editors see that there is a potential audience for your topic of research.

**Response:**

Thank you for your valuable suggestion. To ensure that the cited literature reflects the most recent advances in landslide susceptibility research, we have carefully reviewed and updated the references. Older citations have been removed and replaced with more recent and relevant studies to strengthen the scientific foundation of the manuscript(Delete the redundant literature:"Fell et al., 2008; Hürlimann et al., 2008; Hu & Bentler, 1999; Goren et al., 2010; Manzella et al., 2008."). The revisions have been made in the Introduction (line 82) and Methods Section 2.3.2 of the revised manuscript, as shown below.

**Introduction(Revised manuscript line 82)**

"*Substantial efforts have been made to assess landslide susceptibility using various methodologies, including geoscience factor weighting, statistical models, machine learning, and Geographic Information Systems (GIS)-based spatial analysis (Abay et al., 2019; Gebrehiwot et al., 2025; Guo et al., 2023; Pham et al., 2018; Sun et al., 2024; Wang et al., 2024).*"

**2.3.2 Rationality validation of susceptibility assessment results (Revised manuscript line 246)**

*"In this study, Receiver Operating Characteristic (ROC) curves and area under the curve (AUC) values were used for validation. ROC curves provide a representation of the specificity and sensitivity of an analytical method (Khosravi et al., 2019; Gebrehiwot, et al., 2025). The AUC measures model accuracy, ranging from 0.5 to 1, with values closer to 1 indicating higher accuracy (Wendim et al., 2025)."*

**References:**

Abay, A., Barbieri, G., & Woldearegay, K. (2019). GIS-based Landslide Susceptibility Evaluation Using Analytical Hierarchy Process (AHP) Approach: The Case of Tarmaber District, Ethiopia. *Momona Ethiopian Journal of Science*, 11(1), 14–36. https://doi.org/10.4314/mejs.v11i1

Gebrehiwot, A., Berhane, G., Kide, Y. *et al.* Landslide susceptibility mapping in Lesalso (Laelay Maichew), Northern Ethiopia: a GIS approach using frequency ratio and analytical hierarchy process methods. *Model. Earth Syst. Environ.* **11**, 421 (2025). https://doi.org/10.1007/s40808-025-02578-7

Wendim, S., Mebrahtu, G. & Woldearegay, K. GIS-based landslide susceptibility mapping using Analytical Hierarchy Process method along Gedo-Dilb asphalt road section, Northern Ethiopia. *Bull Eng Geol Environ* **84**, 440 (2025). https://doi.org/10.1007/s10064-025-04455-0

**[Comment 3]** There is some repetition in the introduction, which can be frustrating for your readers; The last paragraph is so long, please split into two part.

**Response:**

Thank you for this helpful comment. We agree that the original Introduction contained some repetitive descriptions, which may reduce readability, and that the final paragraph was overly long.

In response, we carefully revised the Introduction to eliminate redundant statements and improve conciseness, particularly in the discussion of vegetation-related effects on landslide processes. Overlapping explanations were streamlined or merged to avoid repetition while preserving the necessary scientific context.

In addition, the original final paragraph has been reorganized and split into two shorter paragraphs. One now focuses on summarizing the research background and motivation, while the other clearly presents the study objectives and overall contribution. This restructuring improves readability and allows the logical progression of the Introduction to be more clearly conveyed.

We believe these revisions have enhanced the clarity, conciseness, and overall structure of the Introduction. We thank the reviewer for this constructive suggestion, which has helped improve the quality of the manuscript.

**Introduction(Revised manuscript)**

"*Landslides represent a significant geological hazard in mountainous regions worldwide, causing substantial loss of life, infrastructure damage, and economic disruption (Alvioli et al., 2024; Zhang et al., 2025). In areas with dense vegetation cover, the relationship between vegetation and slope stability is particularly complex and non-linear (Deng et al., 2022; Medina et al., 2021). While vegetation is traditionally regarded as a stabilizing agent through root reinforcement, soil moisture regulation, and erosion control (He et al., 2017; Lan et al., 2020; Rey et al., 2019), shallow landslides frequently occur even in densely vegetated landscapes (Xu et al., 2024). This paradox underscores the dual — and often contradictory — role of vegetation in landslide processes, acting as both a mitigating and a predisposing factor depending on environmental context and trigger conditions.*

*The stabilizing function of vegetation is well-documented. Root systems enhance soil cohesion and shear strength, while canopy and litter layers reduce rainfall impact and surface runoff (Gonzalez-Ollauri & Mickovski, 2016; Murgia et al., 2022; Vergani et al., 2017). However, under certain conditions, vegetation can exacerbate slope instability. The added weight of trees, especially on steep slopes, increases gravitational driving forces (Schmaltz & Mergili, 2018). Vegetation can also alter soil hydrological properties, increasing infiltration and soil moisture content, which in turn reduces effective stress and shear resistance during rainfall events (Qin et al., 2022). Furthermore, wind forces acting on tall vegetation can transmit dynamic loads to the slope, while root wedging in thin soils may promote fracture development (Bordoloi & Ng, 2020; Liu et al., 2020). Rainfall remains the primary trigger of landslides in*

*vegetated areas, as it saturates the soil, elevates pore water pressure, and reduces slope stability (Dhanai et al., 2022; Li et al., 2025). Therefore, landslide initiation in vegetated terrain is not governed by vegetation alone but results from the intricate interplay among vegetation characteristics, rainfall intensity, slope gradient, lithology, and other environmental factors.*

*Substantial efforts have been made to assess landslide susceptibility using various methodologies, including geoscience factor weighting, statistical models, machine learning, and Geographic Information Systems (GIS)-based spatial analysis (Abay et al., 2019; Gebrehiwot et al., 2025; Guo et al., 2023; Pham et al., 2018; Sun et al., 2024; Wang et al., 2024). These approaches have improved our understanding of the spatial distribution of landslides and the relative importance of conditioning factors. However, several critical gaps remain. First, many studies provide qualitative descriptions of factor influences but lack quantitative analysis of spatial correlations and interactive effects among multiple driving factors (Shu et al., 2025; Triplett et al., 2025). Second, while rainfall-landslide relationships have been extensively studied using spatial autocorrelation and clustering techniques (Chen et al., 2024; Liu et al., 2024; Ortiz-Giraldo et al., 2023; Pokharel et al., 2021; Wang et al., 2020), the moderating role of vegetation in these relationships is poorly quantified. Specifically, how vegetation mediates the effects of rainfall, lithology, slope, and wind on slope stability coefficients remains unclear (Lan et al., 2020). Third, most susceptibility models operate at a single spatial scale, either regional/watershed or site-specific, with limited integration across scales. This hampers a holistic understanding of how macro-scale predisposing factors translate into micro-scale failure mechanisms.*

*To address these research gaps, this study investigates the dual-edged role of vegetation in landslide susceptibility by integrating watershed-scale statistical analysis with site-specific geomechanical modeling. We selected the Jinkouhe District in Southwest China—a region with high vegetation cover (≥65.5%) and frequent landslide activity—as our study area. The research aims to (1) Quantify the individual and interactive effects of key environmental factors (rainfall, vegetation, wind speed, slope, lithology, etc.) on landslide susceptibility at the watershed scale using Geodetector and Structural Equation Modeling (SEM). (2) Analyze the mechanical role of vegetation weight and its coupling with rainfall and anthropogenic loading in triggering a typical shallow landslide through slope stability calculations. (3) Integrate*

*findings from both scales to elucidate how vegetation mediates landslide processes under different environmental conditions, thereby providing a multi-scale perspective on its "double-edged sword" function. By bridging macroscopic susceptibility patterns with microscopic failure mechanisms, this study offers novel insights into the complex vegetation–landslide interplay. The results are expected to enhance the accuracy of landslide risk assessments and inform sustainable slope management strategies in densely vegetated mountainous regions."*

**References:**

Abay, A., Barbieri, G., & Woldearegay, K. (2019). GIS-based Landslide Susceptibility Evaluation Using Analytical Hierarchy Process (AHP) Approach: The Case of Tarmaber District, Ethiopia. Momona Ethiopian Journal of Science, 11(1), 14–36. https://doi.org/10.4314/mejs.v11i1.

Alvioli, M., Loche, M., Jacobs, L., Grohmann, C. H., Abraham, M. T., Gupta, K., Satyam, N., Scaringi, G., Bornaetxea, T., Rossi, M., Marchesini, I., Lombardo, L., Moreno, M., Steger, S., Camera, C. A. S., Bajni, G., Samodra, G., Wahyudi, E. E., Susyanto, N., Sinčić, M., Gazibara, S. B., Sirbu, F., Torizin, J., Schüßler, N., Mirus, B. B., Woodard, J. B., Aguilera, H., & Rivera-Rivera, J. (2024). A benchmark dataset and workflow for landslide susceptibility zonation. *Earth-Science Reviews*, 258, 104927. https://doi.org/10.1016/j.earscirev.2024.104927

Bordoloi, S., & Ng, C. W. W. (2020). The effects of vegetation traits and their stability functions in bio-engineered slopes: A perspective review. *Engineering Geology*, 275, 105742. https://doi.org/10.1016/j.enggeo.2020.105742

Chen, C., Liu, Y., Li, Y., & Guo, F. (2024). Mapping landslide susceptibility with the consideration of spatial heterogeneity and factor optimization. *Natural Hazards* http://doi.org/10.1007/s11069-024-06955-w

Deng, J., Ma, C., & Zhang, Y. (2022). Shallow landslide characteristics and its response to vegetation by example of July 2013, extreme rainstorm, Central Loess Plateau, China. *Bulletin of Engineering Geology and the Environment*, 81(3), 100. http://doi.org/10.1007/s10064-022-02606-1

Dhanai, P., Singh, V.P. & Soni, P. Rainfall Triggered Slope Instability Analysis with Changing Climate. *Indian Geotech J* **52**, 477–492 (2022). https://doi.org/10.1007/s40098-021-00581-0

Gebrehiwot, A., Berhane, G., Kide, Y. *et al.* Landslide susceptibility mapping in Lesalso (Laelay Maichew), Northern Ethiopia: a GIS approach using frequency ratio and analytical hierarchy process methods. *Model. Earth Syst. Environ.* **11**,

421 (2025). https://doi.org/10.1007/s40808-025-02578-7

Gonzalez-Ollauri, A., & Mickovski, S. B. (2016). Using the root spread information of pioneer plants to quantify their mitigation potential against shallow landslides and erosion in temperate humid climates. *Ecological Engineering*, 95, 302-315. https://doi.org/10.1016/j.ecoleng.2016.06.028

Guo, Z., Guo, F., Zhang, Y., He, J., Li, G., Yang, Y., & Zhang, X. (2023). A python system for regional landslide susceptibility assessment by integrating machine learning models and its application. *Heliyon*, 9(11) https://doi.org/10.1016/j.heliyon.2023.e21542

He, S., Wang, D., Fang, Y., & Lan, H. (2017). Guidelines for integrating ecological and biological engineering technologies for control of severe erosion in mountainous areas – A case study of the Xiaojiang River Basin, China. *International Soil and Water Conservation Research*, 5(4), 335-344. https://doi.org/10.1016/j.iswcr.2017.05.001

Lan, H., Wang, D., He, S., Fang, Y., Chen, W., Zhao, P., & Qi, Y. (2020). Experimental study on the effects of tree planting on slope stability. *Landslides*, 17(4), 1021-1035. http://doi.org/10.1007/s10346-020-01348-z

Li, Z., Guo, J., Li, T. *et al.* Influence of topography on the fragmentation and mobility of landslides. *Bull Eng Geol Environ* **84**, 73 (2025). https://doi.org/10.1007/s10064-025-04095-4

Liu, W., Yang, Z., & He, S. (2020). Modeling the landslide-generated debris flow from formation to propagation and run-out by considering the effect of vegetation. *Landslides*, 18, 43-58. https://doi.org/10.1007/s10346-020-01478-4

Medina, V., Hürlimann, M., Guo, Z., Lloret, A., & Vaunat, J. (2021). Fast physically-based model for rainfall-induced landslide susceptibility assessment at regional scale. *Catena*, 201, 105213. https://doi.org/10.1016/j.catena.2021.105213

Murgia, I., Giadrossich, F., Mao, Z., Cohen, D., Capra, G. F., & Schwarz, M. (2022). Modeling shallow landslides and root reinforcement: A review. *Ecological Engineering*, 181, 106671. https://doi.org/10.1016/j.ecoleng.2022.106671

Ortiz-Giraldo L, Botero BA and Vega J (2023) An integral assessment of landslide dams generated by the occurrence of rainfall-induced landslide and debris flow hazard chain. *Front. Earth Sci.* 11:1157881. http://doi.org/10.3389/feart.2023.1157881

Pham, B. T., Prakash, I., & Tien Bui, D. (2018). Spatial prediction of landslides using a hybrid machine learning approach based on Random Subspace and Classification and Regression Trees. *Geomorphology*, 303, 256-270. https://doi.org/10.1016/j.geomorph.2017.12.008

Pokharel, B., Althuwaynee, O. F., Aydda, A., Kim, S., Lim, S., & Park, H. (2021). Spatial clustering and modelling for landslide susceptibility mapping in the north of the Kathmandu Valley, Nepal. *Landslides*, 18(4), 1403-1419. http://doi.org/10.1007/s10346-020-01558-5

Qin, M., Cui, P., Jiang, Y. et al. Occurrence of shallow landslides triggered by increased hydraulic conductivity due to tree roots. Landslides 19, 2593–2604 (2022). https://doi.org/10.1007/s10346-022-01921-8

Rey, F., Bifulco, C., Bischetti, G. B., Bourrier, F., De Cesare, G., Florineth, F., Graf, F., Marden, M., Mickovski, S. B., Phillips, C., Peklo, K., Poesen, J., Polster, D., Preti, F., Rauch, H. P., Raymond, P., Sangalli, P., Tardio, G., & Stokes, A. (2019). Soil and water bioengineering: Practice and research needs for reconciling natural hazard control and ecological restoration. *Science of the Total Environment*, 648, 1210-1218. http://doi.org/https://doi.org/10.1016/j.scitotenv.2018.08.217

Schmaltz, E. M., & Mergili, M. (2018). Integration of root systems into a GIS-based slip surface model: computational experiments in a generic hillslope environment. *Landslides*, 15(8), 1561-1575. http://doi.org/10.1007/s10346-018-0970-8

Shu, H., Qi, S., Liu, X., Shao, X., Wang, X., Sun, D., ... & He, J. (2025). Relationship between continuous or discontinuous of controlling factors and landslide susceptibility in the high-cold mountainous areas, China. Ecological Indicators, 172, 113313. https://doi.org/10.1016/j.ecolind.2025.113313

Sun, D., Wang, J., Wen, H., Ding, Y., & Mi, C. (2024). Landslide susceptibility mapping (LSM) based on different boosting and hyperparameter optimization algorithms: A case of Wanzhou District, China. *Journal of Rock Mechanics and Geotechnical Engineering*, 16(8), 3221-3232. https://doi.org/10.1016/j.jrmge.2023.09.037

Triplett, L.D., Hammer, M.N., DeLong, S.B. et al. Factors influencing landslide occurrence in low-relief formerly glaciated landscapes: landslide inventory and susceptibility analysis in Minnesota, USA. Nat Hazards 121, 11799–11827 (2025). https://doi.org/10.1007/s11069-025-07262-8

Vergani, C., Giadrossich, F., Buckley, P., Conedera, M., Pividori, M., Salbitano, F., Rauch, H. S., Lovreglio, R., & Schwarz, M. (2017). Root reinforcement dynamics of European coppice woodlands and their effect on shallow landslides: A review. *Earth-Science Reviews*, 167, 88-102. http://doi.org/https://doi.org/10.1016/j.earscirev.2017.02.002

Wang, Y., Feng, L., Li, S., Ren, F., & Du, Q. (2020). A hybrid model considering spatial heterogeneity for landslide susceptibility mapping in Zhejiang Province, China. *Catena*, 188, 104425. http://doi.org/https://doi.org/10.1016/j.catena.2019.104425

Wang, Y., Ling, Y., Chan, T. O., & Awange, J. (2024). High-resolution earthquakeinduced landslide hazard assessment in Southwest China through frequency ratio analysis and LightGBM. *International Journal of Applied Earth Observation and Geoinformation*, 131, 103947. https://doi.org/10.1016/j.jag.2024.103947

Xu, Y., Luo, L., Guo, W., Jin, Z., Tian, P., & Wang, W. (2024). Revegetation Changes Main Erosion Type on the Gully–Slope on the Chinese Loess Plateau Under Extreme Rainfall: Reducing Gully Erosion and Promoting Shallow Landslides. *Water Resources Research*, 60(3), e2023WR036307. https://doi.org/10.1029/2023WR036307

Zhang, Y., Li, y., Tom Dijkstra., Janusz Wasowski., Meng, X., Wu, X., Liu, W., Chen, G. (2025). Evolution of large landslides in tectonically active regions - A decade of observations in the Zhouqu County, China. Engineering Geology, 348, 107967. https://doi.org/10.1016/j.enggeo.2025.107967

**[Comment 4]** Line 67, "reposted" should be "reported".

**Response:** Thank you for pointing out this inappropriate expression. It has been corrected in the revised manuscript.

**[Comment 5]** Figures 1,2 and 4 were referenced in the text after the appearance of the figure. Please amend;

**Response:** Thank you for pointing out the issue regarding the citation order of Figures 1, 2, and 4. We have adjusted the paragraph structure and the placement of the figures accordingly in the revised manuscript.

**[Comment 6]** Clarify why the specific study area (Jinkouhe District) was chosen—how do its characteristics contribute to the relevance of this research?

**Response:**

Thank you for your valuable comment regarding the selection of the study area. The Jinkouhe area was chosen based on the following scientific considerations:

1. High vegetation coverage and complex landslide mechanisms

The Jinkouhe area has a high vegetation coverage (≥65.5%) yet frequently experiences shallow landslides, indicating that the effects of vegetation on landslides are complex and not unidirectionally stabilizing (He et al., 2017; Xu et al., 2024). Previous studies have shown that while vegetation can reduce soil erosion and

enhance slope stability, it may also increase landslide susceptibility due to tree weight or changes in soil properties (Lan et al., 2020; Qin et al., 2024). Therefore, this area provides an ideal setting to investigate the interactive effects of vegetation with rainfall, slope gradient, lithology, and wind forces on landslide susceptibility, thereby revealing the "double-edged sword" role of vegetation.

2. Feasibility of multi-scale analysis

The region's complex topography, diverse geology, and variable hydrological and climatic conditions make it highly suitable for coupled watershed- and point-scale analyses. By applying structural equation modeling (SEM), the Geodetector method, and slope stability coefficient calculations, the influences of individual factors and their interactions on landslide occurrence can be quantified, providing insights into the concealed mechanisms of landslides in highly vegetated areas.

3. Scientific significance

Selecting the Jinkouhe area not only facilitates the investigation of complex landslide mechanisms in regions with dense vegetation but also provides theoretical reference and practical experience for landslide risk assessment and disaster prevention in similar ecological settings. This contributes significantly to understanding the dual role of vegetation in landslide control and the interactive effects of multiple environmental factors.

In addition, to highlight the scientific rationale for this selection, we have supplemented the *Study Area* section with the following statement.

**2.1 Study Area (Revised manuscript line 120)**

"*It provides a representative setting for investigating the dual role of vegetation in landslide occurrence and the coupled influences of multiple environmental factors in highly vegetated mountainous terrains.*"

**References:**

He, S., Wang, D., Fang, Y., & Lan, H. (2017). Guidelines for integrating ecological and biological engineering technologies for control of severe erosion in mountainous

areas – A case study of the Xiaojiang River Basin, China. *International Soil and Water Conservation Research*, 5(4), 335-344. https://doi.org/10.1016/j.iswcr.2017.05.001

Lan, H., Wang, D., He, S., Fang, Y., Chen, W., Zhao, P., & Qi, Y. (2020). Experimental study on the effects of tree planting on slope stability. *Landslides*, 17(4), 1021-1035. http://doi.org/10.1007/s10346-020-01348-z

Qin, M., Cui, P., Jiang, Y. et al. Occurrence of shallow landslides triggered by increased hydraulic conductivity due to tree roots. *Landslides* 19, 2593–2604 (2022). https://doi.org/10.1007/s10346-022-01921-8

Xu, Y., Luo, L., Guo, W., Jin, Z., Tian, P., & Wang, W. (2024). Revegetation Changes Main Erosion Type on the Gully–Slope on the Chinese Loess Plateau Under Extreme Rainfall: Reducing Gully Erosion and Promoting Shallow Landslides. *Water Resources Research*, 60(3), e2023WR036307. https://doi.org/10.1029/2023WR036307

**[Comment 7]** Provide more detailed descriptions of the modeling processes, especially SEM, including assumptions made during factor selection.

**Response:**

Thank you for your insightful comment. We have expanded the description of the Structural Equation Model (SEM) to provide a clearer explanation of the modeling process, including the assumptions underlying factor selection and the relationships among key variables.

**2.4 (2) Structural equation model (Revised manuscript line 303)**

"*The SEM comprises two components: the measurement model, which defines relationships between observed and latent variables, and the structural model, which illustrates relationships among latent variables (Fan et al., 2016; Wang & Rhemtulla, 2021). Based on the GeoDetector results and previous research findings (Chicas et al., 2024; Pourghasemi et al., 2018; Segoni et al., 2024), key factors representing topographic, hydrological, and environmental characteristics were selected to capture the main drivers of landslide susceptibility. The selection of these factors was guided*

*by the assumption that each variable has a direct or indirect physical relationship with landslide occurrence, possesses sufficient explanatory power in the GeoDetector analysis, and reflects geomorphological and ecological processes under high-vegetation conditions. Accordingly, the following hypotheses were proposed:*"

**References:**

Chicas, S. D., Li, H., Mizoue, N., Ota, T., Du, Y., & Somogyvári, M. (2024). Landslide susceptibility mapping core-base factors and models' performance variability: a systematic review. *Natural Hazards*, 120(14), 12573-12593. http://doi.org/10.1007/s11069-024-06697-9

Fan, Y., Chen, J., Shirkey, G., John, R., Wu, S. R., Park, H., & Shao, C. (2016). Applications of structural equation modeling (SEM) in ecological studies: an updated review. *Ecological Processes*, 5(1), 19. http://doi.org/10.1186/s13717-016-0063-3

Pourghasemi, H. R., Teimoori Yansari, Z., Panagos, P., & Pradhan, B. (2018). Analysis and evaluation of landslide susceptibility: a review on articles published during 2005–2016 (periods of 2005–2012 and 2013–2016). *Arabian Journal of Geosciences*, 11(9), 193. http://doi.org/10.1007/s12517-018-3531-5

Segoni, S., Ajin, R. S., Nocentini, N., & Fanti, R. (2024). Insights Gained from the Review of Landslide Susceptibility Assessment Studies in Italy. *Remote Sensing*, *16*(23), 4491. https://doi.org/10.3390/rs16234491

Wang, Y. A., & Rhemtulla, M. (2021). Power Analysis for Parameter Estimation in Structural Equation Modeling: A Discussion and Tutorial. *Advances in Methods and Practices in Psychological Science*, 4(1), 1403230957. http://doi.org/10.1177/2515245920918253

**[Comment 8]** please redraw the Fig. 6.

**Response:**

Thank you for your suggestion.   We have redrawn and optimized Figure 6 by adding a sloping background, illustrating the distribution of shrubs, trees, and grasses

on the slope, and incorporating a rainfall scenario at the top of the figure. These elements were integrated with the original schematic to more clearly depict the conceptual processes, resulting in the revised Figure 6 presented in the manuscript.

[Figure]

*Fig.6. Slope Stress Analysis Diagram*

Furthermore, to comply with the journal's publication requirements, we have also refined several other figures to achieve a consistent style and improved graphical quality across the manuscript. For Figure 1, we added a new basemap for panels (a) and (b). In (a), the names of other provinces were removed, while Sichuan Province was retained and highlighted. Panel (b) now more clearly shows the location of the study area. In (c), the color scheme and north arrow were updated for better visual clarity. In addition, panels (d–h) were supplemented with partially interpreted landslides in densely vegetated areas based on visual interpretation.

[Figure]

*Fig. 1. Location map of Jinkouhe District*

For **Figure 7**, the north arrow was replaced, and a new layer title "LSI" was added to improve the figure's interpretability.

[Figure]

*Fig. 2. Landslide susceptibility assessment map*

For **Figure 9**, the image was simplified by replacing the previously cluttered multicolor layout with a yellow-and-blue scheme. Yellow arrows represent the total effects of conditioning factors on landslide susceptibility, while blue dashed lines indicate the indirect interactions among factors.

[Figure]

*Fig. 3. SEM of landslide susceptibility.*

For **Figure 10**, we removed the landslide point distribution to make the map clearer, and added a legend, scale bar, and north arrow for each scenario to improve consistency and clarity.

[Figure]

*Fig. 4. Landslide Susceptibility Distribution Map for Five Scenarios*

**[Comment 9]** Lines 330-332, It would be useful to give the areas (north, southwest, etc.);

**Response:**

Thank you for your insightful comment. We have adjusted the sentence order in this section and re-added directional information to clarify the distribution of high and very high susceptibility zones. The revised text is as follows:

**3.1 Landslide susceptibility mapping and distribution characteristics(Revised manuscript line 369)**

"*Moderate susceptibility zones are widespread across the northern, western, and southwestern regions. High and very high susceptibility zones, though smaller in coverage, exhibit a "cross-shaped" spatial distribution, primarily located in the central-eastern and northeastern parts of the study area, with a small portion in the southwest. Very high susceptibility zones are scattered within the high susceptibility areas.*"

**[Comment 10]** Line 335, please left a space between Fig.7.landslide……

**Response:** Thank you for pointing out this error. We have corrected it in the revised manuscript.

**[Comment 11]** Lines 352-353, the title 3.3 can be revised as "Slope stability calculation considering artificial waste sediment and vegetation self-weight".

**Response:** Thank you for your precise comment. We agree with your suggestion and have revised the original subsection title accordingly.

**[Comment 12]** This section provides a thorough discussion of the results-based part and the differences and improvements compared to previous studies, and also offers an outlook on future work. However, it is necessary to more clearly point out the unique aspects of this research (for example, "integrating macroscopic susceptibility with microscopic mechanics"). The outlook for future research can be more specific, for example: how to utilize interpretable machine learning and multi-source data fusion, rather than just making general statements:

**Response:** Thank you very much for this valuable and constructive comment. We appreciate the reviewer's recognition of the comprehensive discussion of our results and agree that the unique aspects and future perspectives of this study should be stated more clearly. In the revised manuscript, we emphasize that the novelty of this research lies in revealing that even areas with dense vegetation coverage, which are often considered stable, can still experience shallow landslides under the combined influence of rainfall, vegetation weight, and human disturbances. By integrating macroscopic susceptibility analysis with microscopic mechanical interpretation, our study connects regional-scale assessments with field-scale processes and provides new insight into the dual role of vegetation in slope stability.

In addition, the Outlook section has been refined to include more targeted content. Future research could consider applying optical remote sensing image classification and InSAR deformation monitoring to identify potentially unstable slopes. At the same time, interpretable machine learning models, such as SHAP-based approaches, could be used to quantify the nonlinear interactions, threshold effects, and spatial heterogeneity among key conditioning factors. These methods would improve the interpretability of susceptibility assessments and enhance the temporal and spatial resolution of landslide prediction and early warning in densely vegetated mountainous regions. These additions make the outlook more specific and provide feasible directions for extending the current research.

**4.4 Comparison with previous studies and scope for future research(Revised manuscript line 593)**

"*Existing studies on landslides have predominantly focused on rainfall-related triggering mechanisms, such as rainfall intensity, duration, and antecedent moisture conditions (Gatto et al., 2025; Zhang et al., 2025). For example, Cui et al. (2024) analyzed the characteristics and causes of a similar landslide in this area using Massflow V2.8 simulations. They identified rainfall and human activities as key triggers, but insufficiently addressed interactions between soil, moisture, and external forces (such as natural wind and human mining activities) under high vegetation*

conditions. This limited simulation accuracy. In these studies, vegetation is often treated as a background environmental condition or a stabilizing factor, while its mechanical and hydrological roles are rarely quantified explicitly. As a result, landslides occurring in highly vegetated areas are commonly interpreted primarily as a response to extreme rainfall, with comparatively limited attention paid to vegetation-related processes themselves. Consequently, from the perspective of vegetation as an active influencing factor, research addressing why landslides still occur in areas with dense vegetation coverage remains relatively scarce.

Furthermore, An et al. (2025) investigated the mechanisms of landslide occurrence in densely vegetated areas by examining the interactions between terrain and lithological properties. They highlighted that in natural forests, landslides tend to initiate along the soil$-$bedrock interface. Owing to the shallow soil layer and pronounced permeability contrast, perched water readily accumulates above this interface, thereby reducing shear strength and triggering slope failure. Their work underscores the significant role of vegetation as a key intermediary that links various environmental factors in shaping landslide susceptibility. Nevertheless, their study treated terrain and lithology primarily as background environmental conditions and did not account for slope damage induced by wind drag on trees. In contrast, the present study incorporates wind forces both in the macroscopic assessment of landslide susceptibility and in the stability analysis of specific slopes.The results of our study support and extend these findings by demonstrating that high vegetation coverage does not necessarily imply low landslide susceptibility.

Our study integrates regional-scale susceptibility assessment with site-scale mechanical interpretation. This multi-scale framework bridges macroscopic statistical patterns and microscopic physical processes, providing a more comprehensive understanding of vegetation's "double-edged" effect on landslide development. The novelty of this work lies not only in identifying the limitations of vegetation's stabilizing role, but also in clarifying the conditions under which its negative effects may become significant. But research on vegetation types, height, and growth conditions (such as thickness and types of soil and human activity disturbances) in relation to landslide risks remains limited. Future research could apply optical remote sensing image classification and InSAR deformation monitoring to identify potentially unstable slopes and capture temporal deformation characteristics (Li et al., 2025). When combined

*with interpretable machine learning approaches, such as SHAP-based models, together with analytical tools like GeoDetector and SEM, these methods can quantify nonlinear interactions, threshold effects, and spatial heterogeneity among conditioning factors (Sun et al., 2024; Wen et al., 2025), thereby improving the interpretability of susceptibility evaluation and enhancing the prediction capability for landslides in densely vegetated areas.*"

**References:**

An, N., Dai, Z., Hu, X., Wang, B., Huo, Z., Xiao, Y., ... & Yang, Y. (2025). Comparing landslide patterns and failure mechanisms in restored and native forest ecosystems: Insights from geomorphology, lithology and vegetation. Catena, 260, 109452. https://doi.org/10.1016/j.catena.2025.109452

Cui, Y., Qian, Z., Xu, W., & Xu, C. (2024). A Small-Scale Landslide in 2023, Leshan, China: Basic Characteristics, Kinematic Process and Cause Analysis. Remote Sensing, 16(17), 3324. http://doi.org/10.3390/rs16173324

Gatto M P A, Misiano S, Montrasio L. Space-time prediction of rainfall-induced shallow landslides through Artificial Neural Networks in comparison with the SLIP model[J]. Engineering Geology, 2025, 344: 107822. https://doi.org/10.1016/j.enggeo.2024.107822

Li, Q., Li, X., Zhao, C., & Zhang, S. (2025). Back analysis key parameters of Scoops3D model using SBAS-InSAR technology for regional landslide hazard assessment. Landslides, 22(12), 4097-4112. https://doi.org/10.1007/s10346-025-02578-9

Sun, D., Wang, J., Wen, H., Ding, Y., & Mi, C. (2024). Landslide susceptibility mapping (LSM) based on different boosting and hyperparameter optimization algorithms: A case of Wanzhou District, China. Journal of Rock Mechanics and Geotechnical Engineering, 16(8), 3221-3232. https://doi.org/10.1016/j.jrmge.2023.09.037

Wen, H., Yan, F., Huang, J., & Li, Y. (2025). Interpretable machine learning models and decision-making mechanisms for landslide hazard assessment under different rainfall conditions. Expert Systems with Applications, 270, 126582. https://doi.org/10.1016/j.eswa.2025.126582

Zhang L M, Lü Q, Deng Z H, et al. Probabilistic approach to determine rainfall thresholds for rainstorm-induced shallow landslides using long-term local precipitation records[J]. Engineering Geology, 2025: 108139. https://doi.org/10.1016/j.enggeo.2025.108139

**[Comment 13]** Line 364, the title should be changed to "Analysis of landslide driving

factors and their interaction pathways". This part, the authors mainly emphasize the factors and the interactions.

**Response:** Thank you for your valuable comment. We agree with your suggestion and have revised the subsection title accordingly.

**[Comment 14]** Line 453, the title should be precise. This part mainly compared the landslide susceptibility under different factors combination, so maybe this title will be more suitable: "differences and explanations of landslide susceptibility results under different factor combinations".

**Response:** Thank you for pointing out this inappropriate expression. It has been corrected in the revised manuscript.

Thank you again for the reviewer's constructive comments. We hope the revisions and responses will make our manuscript suitable for publication in ***Natural Hazards and Earth System Sciences.***

---

## Author Comment (AC2)

**Response letter**

**Dear Editors and Reviewers:**

**Re: egusphere-2025-3004**

We sincerely thank you and reviewers for providing us with such a valuable revision opportunity. Thus, we can further improve and present our studies. The comments from you and the reviewers were highly insightful and enabled us to greatly improve the quality of our manuscript. We have carefully reviewed the feedback and made corrections that we hope will be met with approval. Revised portions are marked on the revised manuscript. Please note that these resulting revisions did not change the paper's findings.

In the response letter to editor and reviewers, we firstly summarized the major changes in a cover letter to editors, and we then itemized response to editors and reviewers, **in which the blue font indicates the response to each comment and the black font presents the revision from the revised manuscript.**

We hope that the revisions in the revised manuscript and the responses to the comments will suffice to allow our manuscript to be suitable for publication in *Natural Hazards and Earth System Sciences*.

**Sincerely regards,**

**Songtang He (hest@imde.ac.cn)**

**Institute of Mountain Hazards and Environment, Chinese Academy of Science**

**Response to Reviewer #2**

**[Comment 1]** General structure and integration between scales

A central weakness of the paper lies in the lack of clarity regarding how the local-scale analyses are integrated with the regional-scale susceptibility assessment. As currently presented, the local-scale analyses appear disconnected and scientifically irrelevant, adding little to the main argument. If the authors cannot clearly demonstrate the conceptual and methodological linkage between the two scales, I would recommend removing the local-scale component entirely.

**Response:**

Thank you very much for this valuable comment. We agree that the connection between the major landslide case and the large-scale susceptibility analysis was not clearly presented in the original manuscript. The purpose of introducing this landslide is to use it as a representative case study that links the regional-scale susceptibility assessment with the site-scale mechanisms of slope failure, providing a field-based context for subsequent susceptibility analysis and mechanistic interpretation. The landslide was triggered by the combined effects of prolonged rainfall, anthropogenic loading from waste deposits, and the additional weight of dense vegetation. This event highlights the amplifying effect of the interaction between vegetation and rainfall, indicating that local environmental disturbances can significantly increase landslide risk. In other words, vegetation may enhance slope stability under certain conditions but can also aggravate slope failure due to its additional weight and water-retention capacity. Therefore, this case not only provides empirical validation for the regional analysis results but also reveals the amplification of large-scale controlling factors under local conditions, further supporting the "double-edged sword" role of vegetation identified through the GeoDetector and SEM analyses.

We have revised the manuscript accordingly. Specifically, we have clarified the conceptual linkage between scales at the end of the Introduction, strengthened the role of the case study in the Study Area description, and explicitly integrated the localscale findings into the Discussion to demonstrate their relevance to the regional susceptibility assessment.

**1 Introduction(Revised manuscript line 100)**

"To address these research gaps, this study investigates the dual-edged role of vegetation in landslide susceptibility by integrating watershed-scale statistical analysis with site-specific geomechanical modeling. We selected the Jinkouhe District in Southwest China—a region with high vegetation cover (≥65.5%) and frequent landslide activity—as our study area. The research aims to (1) Quantify the individual and interactive effects of key environmental factors (rainfall, vegetation, wind speed, slope, lithology, etc.) on landslide susceptibility at the watershed scale using Geodetector and Structural Equation Modeling (SEM). (2) Analyze the mechanical role of vegetation weight and its coupling with rainfall and anthropogenic loading in triggering a typical shallow landslide through slope stability calculations. (3) Integrate findings from both scales to elucidate how vegetation mediates landslide processes under different environmental conditions, thereby providing a multi-scale perspective on its "double-edged sword" function. By bridging macroscopic susceptibility patterns with microscopic failure mechanisms, this study offers novel insights into the complex vegetation–landslide interplay. The results are expected to enhance the accuracy of landslide risk assessments and inform sustainable slope management strategies in densely vegetated mountainous regions."

**2.1 Study area(Revised manuscript line 142)**

"This study takes the JinKouhe area as the research focus. A major landslide occurred in the study area on June 4, 2023, near the living quarters of a Phosphate Mine (103°2'25.75" E, 29°25'0.6" N), which serves as a representative case providing field evidence for the subsequent discussion of slope-failure mechanisms."

**4.3 Mechanisms of landslides in areas with high vegetation coverage(Revised manuscript line 544)**

"At the watershed scale, the GeoDetector results indicate that NDVI alone exhibits limited independent explanatory power (q = 0.27, Table 4). However, its

interaction with rainfall significantly enhances landslide susceptibility (e.g., NDVI $\times$ rainfall q = 0.67, Fig. 8), suggesting that vegetation can amplify the destabilizing effects of precipitation under certain conditions. While vegetation intercepts rainfall and promotes evapotranspiration, it can also alter soil moisture distribution via stemflow, root-induced preferential flow, and reduced surface runoff. Under prolonged rainfall, these processes may lead to localized saturation, thereby exacerbating landslide and debris flow risks in vegetated slopes. This aligns with the SEM results, which attribute a total indirect effect of 0.21 to NDVI, mediated largely through soil moisture dynamics and interactions with rainfall and slope (Fig. 9, Table 5). The susceptibility scenario analysis further illustrates this duality: adding vegetation alone (Class II) slightly reduced the extent of very high susceptibility zones, yet when combined with rainfall (Class IV) and wind (Class V), it led to a notable expansion of high-susceptibility areas and an increase in landslide counts (Table 6, Fig. 10). This suggests that vegetation's protective capacity may be offset or reversed under prolonged rainfall, especially on steeper slopes.

At the site-specific scale, the stability calculations provide direct mechanical insight into how vegetation can transition from a stabilizing to a destabilizing factor. Under natural (unsaturated) conditions, the slope remained stable even with the added weight of vegetation and waste material (Fs = 1.02). However, under saturated conditions, the same additional loads—particularly the self-weight of trees—reduced the stability coefficient to 0.89, triggering failure (Table 3). This demonstrates that the mechanical reinforcement from roots can be outweighed by the gravitational load of vegetation when soil strength is reduced by saturation, a shift that is quantitatively captured by our modeling.

These findings help explain why landslides may occur unexpectedly in densely vegetated areas. Vegetation can create a false sense of stability by masking early signs of movement (e.g., surface cracking, minor slumping) and by being traditionally associated with slope protection.  Moreover, the same root networks that enhance soil cohesion also facilitate preferential infiltration, potentially accelerating soil saturation during heavy rainfall—a process reflected in the strong interaction between NDVI and rainfall in our spatial analysis. In terrain with high lateral variability in slope, lithology, or soil depth, vegetation may thus contribute to highly localized and

*concealed instability, as exemplified by the 2023 Jinkouhe landslide.*"

**[Comment 2]** Methodological adequacy – Use of AHP: The use of the Analytic Hierarchy Process (AHP) as the core method for landslide susceptibility mapping raises serious concerns. AHP is highly subjective and largely outdated, having been replaced in the literature by more objective and data-driven approaches (e.g., statistical models, machine learning algorithms, ensemble frameworks). The authors do not provide any convincing justification for this methodological choice. In its current form, this decision undermines the robustness and reproducibility of the results.

**Response:**

Thank you for your insightful comment. We fully understand your concern regarding the use of the Analytic Hierarchy Process (AHP). To clarify, AHP is the primary method used in the manuscript for deriving factor weights and producing the susceptibility map. AHP was selected for its transparent and systematic weighting mechanism that facilitates the explicit incorporation of expert judgment, making it well suited for regions characterized by limited or uneven landslide inventory data. We do not claim that AHP is superior to data-driven techniques; rather, in this work AHP serves as the main, interpretable mapping approach appropriate for the study objectives and available data.

Moreover, AHP has continued to be widely applied in recent years for landslide susceptibility mapping due to its interpretability and ease of factor weighting (e.g., Alamrew et al., 2024; Asmare, 2023; Gebrehiwot et al., 2025; Liu et al., 2024; Mustapha et al., 2025). To further address your concern, we conducted a supplementary validation experiment using four interpretable machine learning models—XGBoost, LightGBM, GBDT, and CatBoost—to evaluate the robustness and reproducibility of our AHP-based results. The workflow was as follows: (1) all conditioning factors were extracted and divided into two categories (landslide and non-landslide samples); (2) data were standardized and randomly split into training (70%) and testing (30%) subsets; (3) the Optuna heuristic optimization framework

was employed to tune model hyperparameters, replacing the traditional grid search approach; and (4) model performance was compared using ROC curves (Figure R1). The testing AUC values were 0.816 (CatBoost), 0.784 (GBDT), 0.773 (LightGBM), and 0.766 (XGBoost).

[Figure]

*Fig. R1.* **ROC curves comparing the four machine learning models**

Furthermore, we applied SHAP analysis to the CatBoost model to interpret feature contributions and found that the main controlling factors identified, such as Elevation, Slope, Distance to fault and Mean annual rainfall, were largely consistent with those derived from the GeoDetector analysis. The slight discrepancies are reasonable since SHAP evaluates the influence of each variable within the model's decision process, while GeoDetector emphasizes spatial heterogeneity.

[Figure]

*Fig. R2. SHAP summary (swarm) plot for the CatBoost model*

*Fig. R3. Mean absolute SHAP values for the CatBoost model*

Overall, the AHP model exhibited comparable predictive performance and similar dominant factors to the machine learning models, confirming the robustness and reproducibility of our results. More importantly, the negative samples in this supplementary experiment were generated using the same buffer-based method as in the AHP validation. This ensures the comparability of the results. Although the AUC values of the machine learning models are moderate, their performance could be further enhanced by incorporating more refined sampling strategies, such as factor-based spatial optimization or model-driven negative sample optimization frameworks. These improvements will be considered in our future work to enhance the precision and generalizability of machine learning–based susceptibility models.

**References**

Alamrew, B. T., Kassawmar, T., Mengstie, L., & Jothimani, M. (2024). Combined GIS, FR and AHP approaches to landslide susceptibility and risk zonation in the Baso Liben district, Northwestern Ethiopia. *Quaternary Science Advances*, *16*, 100250. https://doi.org/10.1016/j.qsa.2024.100250

Asmare, D. (2023). Application and validation of AHP and FR methods for landslide susceptibility mapping around choke mountain, northwestern ethiopia. Scientific African, 19, e1470. https://doi.org/10.1016/j.sciaf.2022.e01470

Gebrehiwot, A., Berhane, G., Kide, Y. et al. Landslide susceptibility mapping in Lesalso (Laelay Maichew), Northern Ethiopia: a GIS approach using frequency ratio and analytical hierarchy process methods. Model. Earth Syst. Environ. 11, 421 (2025). https://doi.org/10.1007/s40808-025-02578-7

Liu, X., Shao, S., & Shao, S. (2024). Landslide susceptibility zonation using the analytical hierarchy process (AHP) in the Great Xi'an Region, China. Scientific Reports, 14(1), 2941. http://doi.org/10.1038/s41598-024-53630-y

Mustapha, A. I. T., Etebaai, I., Taher, M., & Tawfik, A. (2025). Landslide Susceptibility Mapping in the Bokoya Massif, Northern Morocco: A Geospatial and Multi-Factor Analysis Using the Analytic Hierarchy Process (AHP). *Scientific African*, e02980. https://doi.org/10.1016/j.sciaf.2025.e02980

**[Comment 3]** Landslide inventory and model validation: The description of the landslide inventory is inconsistent and insufficient. The manuscript states that the inventory was downloaded from an online repository and then integrated with 227 manually identified landslides; yet later it is mentioned that the total number of landslides is 227. This discrepancy must be clarified. The authors should:

(i) provide a detailed and transparent description of the inventory, including sources, validation, and completeness; and

(ii) clearly explain the selection criteria for non-landslide points, as this strongly affects the ROC/AUC results.

Without this information, the reported validation accuracy appears potentially overestimated and unreliable.

**Response:**

We sincerely thank you for the valuable comments regarding the landslide inventory and model validation. The manuscript has been revised to provide a clearer and more detailed description, and we hope these clarifications address your concerns.

1. In this study, the landslide inventory was compiled from two sources: the first

source was the landslide inventory of the Jinkouhe area provided by the GeoCloud platform, and the second source consisted of landslides identified through visual interpretation of high-resolution satellite imagery (Sentinel-2) and manual verification. The datasets were integrated, and duplicate or uncertain cases were removed, resulting in a total of 227 landslides, representing the complete landslide distribution within the study area. To address the unclear description in the original manuscript, this has been revised in **# Revised manuscript line 196.** To visually demonstrate the reliability of some landslide points in the study area, Figure R4 shows a subset of landslides identified through visual interpretation, and part of these landslides have been added to Figure 1 in the revised manuscript. And the descriptions are below.

**2.2 Data source and preprocessing(Revised manuscript line 197)**

*"(8) Landslide hazard point data were obtained from the GeoCloud platform (https://geocloud.cgs.gov.cn/) and through manual interpretation of Sentinel-2 imagery acquired in August 2024. After integrating the sources and removing duplicates and uncertain cases, the final inventory consisted of 227 validated landslides, representing the complete distribution within the study area. Some representative landslides identified through visual interpretation are shown in Figure 1d–h."*

[Figure]

*Fig. R4 Visually interpreted subset of landslides*

2. To assess the reliability of the dataset and the robustness of the model, a 500 m buffer was generated around the known landslide points, and areas close to water bodies were excluded. Within the remaining regions, non-landslide points (negative samples) were randomly selected while maintaining a balanced ratio between positive and negative samples. Following the repeated random sampling approach commonly used in machine learning studies, 30% of the entire dataset was repeatedly and randomly selected as the testing subset for AUC validation. Multiple repeated validations were performed, and the resulting AUC values showed minimal variation (<0.05), indicating consistent and reliable model performance. These clarifications have been added in the revised manuscript, line 252.

**2.3.2 Rationality validation of susceptibility assessment results(Revised manuscript line 252)**

"*Based on the susceptibility distribution map and known landslide points, non-landslide points were randomly sampled from areas excluding water bodies and 500 m landslide buffer zones to maintain a balanced ratio between positive and negative samples. Following the repeated random sampling approach commonly used in machine learning studies, 30% of the entire dataset was repeatedly and randomly selected as the testing subset for ROC/AUC evaluation to assess model robustness. The mean AUC values obtained from the five scenarios were 0.774, 0.786, 0.795, 0.810, and 0.801, respectively—all exceeding 0.5. The variation in AUC across repetitions was minimal (<0.05), indicating consistent and reliable landslide susceptibility evaluation (Fig. 4).*"

[Comment 4] Reference to established best practices. The authors should explicitly compare their approach with the guidelines proposed by Reichenbach et al. (2018), who outlined key criteria for producing reliable landslide susceptibility maps. Currently, the manuscript neither demonstrates adherence to these well-established standards nor engages with them critically.

**Response:**

Thank you for your valuable comment. We carefully reviewed the work of Reichenbach et al. (2018) and compared their guidelines with our modeling approach. First, regarding the selection of conditioning factors, section 3.4.1 in Reichenbach's manuscript (Reichenbach et al. 2018) noted that for each susceptibility model, two to twenty-two thematic variables were typically used, with an average of nine. In our study, we selected ten factors—elevation, slope, aspect, distances to faults, rivers, and roads, lithology, rainfall, NDVI, and wind speed—which are sufficient to construct a robust model. Except for wind speed, all factors are commonly used in landslide susceptibility research. Second, in terms of mapping units, we adopted grid cells (30 × 30 m), which correspond to one of the three basic evaluation units recommended by Reichenbach et al. (2018). Finally, for model validation, we used the Receiver Operating Characteristic (ROC) curve, which is also mentioned as common practice in their work. Taken together, we believe that our methodology follows the best-practice guidelines proposed by Reichenbach et al. (2018). Accordingly, we have revised the manuscript to include the following statement at the beginning of Section 2.3.1.

2.3.1landslide susceptibility based on the analytic hierarchy process(Revised manuscript line 207)

"*Following the best-practice guidelines for landslide susceptibility mapping proposed by Reichenbach et al. (2018).*"

**References**

Reichenbach, P., Rossi, M., Malamud, B. D., Mihir, M., & Guzzetti, F. (2018). A review of statistically-based landslide susceptibility models. *Earth-science reviews*, *180*, 60-91. https://doi.org/10.1016/j.earscirev.2018.03.001

**[Comment 5]** Analysis of variables and use of geodetector: The attempt to explore how different variables (and their combinations) influence susceptibility is potentially interesting. However, the application of the GeoDetector method appears

methodologically flawed: the authors use the susceptibility map (derived from AHP) as the dependent variable, rather than the landslide inventory itself. Since susceptibility is already a model — and a highly subjective one — this approach introduces a strong bias, making any subsequent inferences about controlling factors or vegetation effects questionable. Such analyses should be based on observed landslide occurrences, not on the output of another model.

**Response:** Thank you for your valuable comment. We fully understand your concern regarding the GeoDetector method and the choice of dependent variable. In most GeoDetector studies, the spatial distribution of landslide occurrences (i.e., the 0–1 variable representing landslide and non-landslide areas) is commonly used as the dependent variable to identify the major conditioning factors strongly associated with landslide occurrence. However, this approach is more suitable for the preliminary stage of factor screening (Liu et al., 2024; Sun et al., 2021; Yang et al., 2019; Zhou et al., 2023), as it can reveal which environmental variables show strong spatial consistency with landslide occurrences, but cannot effectively reflect how multiple factors interact or jointly influence the degree of landslide susceptibility.

The fundamental reason is that the 0–1 variable only represents two discrete states—"occurrence" or "non-occurrence"—and lacks information about continuous variation. It can only capture the spatial consistency between individual factors and landslide occurrence. In other words, when the dependent variable is binary (0–1), GeoDetector can answer the question "Which factors are correlated with landslide occurrence?" but not "How do these factors jointly shape the spatial distribution and intensity of landslide susceptibility?". Therefore, such analyses reveal the spatial pattern of landslide occurrence rather than the underlying mechanism that controls the spatial heterogeneity of landslide susceptibility.

In this study, we used the landslide susceptibility map derived from the AHP model as the dependent variable, which serves a different purpose from the conventional approach. The susceptibility index is a continuous variable that represents the relative probability of landslide occurrence rather than a simple binary

state of "occurred" or "not occurred." Applying GeoDetector to this continuous susceptibility map allows us to quantitatively evaluate the explanatory power of each factor on spatial variations in susceptibility and further explore how factor interactions jointly influence the susceptibility pattern. This analysis helps clarify which environmental factors and combinations dominate the spatial variation of landslide susceptibility in the study area.

Therefore, this process is not a simple repetition of the AHP model results but rather a mechanism-oriented interpretation and validation from the perspective of spatial heterogeneity. In other words, using the landslide susceptibility map as the dependent variable in GeoDetector is not a methodological error but a deliberate design aimed at shifting the focus from the occurrence of landslide events to understanding the spatial mechanisms that shape landslide susceptibility, thereby giving the AHP results a clearer physical interpretation (Chen et al., 2023). And We fully understand your concern that "the susceptibility map itself is a model output, which may introduce bias." To verify the rationality and reliability of applying the GeoDetector method based on the susceptibility map, we conducted additional experiments and comparative analyses from two complementary perspectives.

(1) GeoDetector comparison based on binary landslide classification

We re-applied GeoDetector using actual landslide and non-landslide samples (0– 1 classification) as the dependent variable with the same set of conditioning factors. The results showed that the major explanatory factors—such as slope, distance to fault, and mean annual rainfall—were largely consistent with those identified by the AHP-GeoDetector analysis (Figure R5), with only minor differences in secondary factors. This demonstrates that the relative importance of conditioning factors remains stable even when the dependent variable differs, confirming the robustness and interpretability of the AHP-GeoDetector results.

[Figure]

*Fig. R5 GeoDetector factor importance using binary (0–1) landslide classification.*

(2) Independent validation using a machine learning model

We also performed SHAP-based interpretability analysis using the CatBoost model from our previous supplementary experiments and compared its results with the AHP-GeoDetector analysis (Comment 2, Figure R2&3). The findings showed a high degree of agreement: key controlling factors such as elevation, slope, distance to fault, and mean annual rainfall were consistently identified as dominant contributors. This cross-method consistency indicates that the identification of landslide-driving factors is reproducible and stable across different modeling frameworks, demonstrating that the AHP results were not artifacts of model subjectivity.

In summary, our comparative analyses empirically validate the methodological soundness of applying GeoDetector to the susceptibility map. Although the AHP is an expert-based weighting model, its outputs show strong consistency and spatial explanatory power when verified by multiple independent approaches. Therefore, the GeoDetector analysis in this study did not introduce significant bias, but rather enhanced the understanding of the spatial mechanisms underlying landslide susceptibility formation.

**References**

Chen, Z., Song, D. & Dong, L. An innovative method for landslide susceptibility mapping supported by fractal theory, GeoDetector, and random forest: a case study in Sichuan Province, SW China. Nat Hazards 118, 2543–2568 (2023). https://doi.org/10.1007/s11069-023-06104-9

Liu, X., Shao, S., & Shao, S. (2024). Landslide susceptibility prediction and mapping in Loess Plateau based on different machine learning algorithms by hybrid factors screening: Case study of Xunyi County, Shaanxi Province, China. Advances in Space Research, 74(1), 192-210. https://doi.org/10.1016/j.asr.2024.03.074

Sun, D., Shi, S., Wen, H., Xu, J., Zhou, X., & Wu, J. (2021). A hybrid optimization method of factor screening predicated on GeoDetector and Random Forest for Landslide Susceptibility Mapping. Geomorphology, 379, 107623. https://doi.org/10.1016/j.geomorph.2021.107623

Yang, J., Song, C., Yang, Y., Xu, C., Guo, F., & Xie, L. (2019). New method for landslide susceptibility mapping supported by spatial logistic regression and GeoDetector: A case study of Duwen Highway Basin, Sichuan Province, China. Geomorphology, 324, 62-71.https://doi.org/10.1016/j.geomorph.2018.09.019

Zhou, X., Wen, H., Zhang, Y., Xu, J., & Zhang, W. (2021). Landslide susceptibility mapping using hybrid random forest with GeoDetector and RFE for factor optimization. Geoscience Frontiers, 12(5), 101211. https://doi.org/10.1016/j.gsf.2021.101211

**[Comment 6]** Structure and clarity of the manuscript: The manuscript is redundant and lacks structural clarity. I recommend:(i) removing repetitive sentences;(ii) improving logical flow and conciseness throughout the text. In particular, several statements about the "ambiguous role" of vegetation are speculative and not sufficiently substantiated by quantitative evidence.

**Response:**

(1) we have carefully revised the manuscript to remove repetitive sentences and improve logical flow and conciseness. Statements related to the role of vegetation have been consolidated, and the end of the Introduction, the Study Area description, and the Discussion section have been reorganized to better integrate the multi-scale analyses.

(2) We thank you for highlighting the need for stronger quantitative support in Section 4.3 regarding the "ambiguous role" of vegetation. We agree that some statements in this subsection appeared speculative, and we have revised Section 4.3 to better integrate our empirical findings and quantitative results. Detailed explanations and corresponding manuscript revisions are provided in our response to Comment 9 (Response 26).

**[Comment 7]** Vegetation: The paper focuses on vegetation but does not include an accurate description of the types of vegetation present in the study area. This is a major issue in my opinion.

**Response:**

Thank you for your valuable comment. As you rightly pointed out, to provide a more accurate description of the vegetation types in the study area, we obtained the vegetation distribution based on the 1:1,000,000 China Vegetation Type Spatial Distribution vector data, as shown in the figure R6. The vegetation in the study area is mainly classified into shrubland, meadow, broadleaf forest, coniferous forest, and cultivated plants. Among them, the shrubland mainly consists of Myrica and Rhododendron; broadleaf forests mainly include Arundinaria-dominated forests, Quercus engleriana forests, and Castanopsis forests; and coniferous forests are primarily composed of Abies forests, Pinus yunnanensis forests, and subalpine Quercus forests. This information has been added to Section 2.1 "Study Area".

**2.1 Study Area(Revised manuscript line 134)**

"*Vegetation is classified into shrubland, meadow, broadleaf forest, coniferous forest, and cultivated plants. Shrubland is dominated by Myrica and Rhododendron,*

*broadleaf forests include Arundinaria-dominated forests, Quercus engleriana forests, and Castanopsis forests, and coniferous forests consist mainly of Abies forests, Pinus yunnanensis forests, and subalpine Quercus forests.*"

[Figure]

*Fig. R6 Vegetation type distribution map*

**[Comment 8]** Specific (but not minor) comments

1. l29: "landslide frequently occur" → How frequently? More than in other soil-cover conditions? I suggest replacing frequently with may.

**Response:**

Thank you for the suggestion. To improve the academic rigor and structural clarity of the Introduction, we substantially revised this section. During the revision, the original sentence containing the expression "frequently occur" was removed and no longer appears in the revised Introduction.

2. l59: What is meant by "good vegetation"?

**Response:**

Thank you for your comment. The original text is "...under good vegetation cover conditions...", which refers to areas with dense or well-developed vegetation cover. According to the suggestion on the introduction revision, I have completely rewritten the introduction section, removing the repetition, some ambiguous words, and logical

errors. Please check the introduction section in the manuscript, lines 55-116, pages 4-6.

3. l60: change to: "mitigation also depends on…"

**Response:**

Thank you for your suggestion. We appreciate your attention to phrasing. To improve the academic rigor and structural clarity of the Introduction, we substantially revised this section. During the revision, the original sentence containing the expression "mitigation also depends on" was removed and no longer appears in the revised Introduction.

4. l73: What do you mean by "susceptibility to external disturbances"? I believe susceptibility refers specifically to landslides in this paper.

**Response:**

Thank you for your comment. In this context, "susceptibility to external disturbances" refers to the slope's sensitivity to external triggering factors such as rainfall, seismic activity, or human disturbance. This phrasing was used to emphasize that root wedging can increase the likelihood of slope instability when exposed to such external factors. In this study, landslide susceptibility is specifically quantified using the Landslide Susceptibility Index (LSI)

5. l75: You probably mean apparent cohesion, since cohesion itself is not reduced by rainfall.

**Response:**

Thank you for this insightful comment. After further verification and consultation with domain experts, we confirm that soil cohesion is a standard technical term widely used in geotechnical literature to represent the cohesive component of soil shear strength in slope stability analyses. In this context, the original expression is scientifically appropriate and has therefore been retained in the revised manuscript.

6. l76: "downslope forces generated by gravitational water distribution" — please clarify.

**Response:**

Thank you for your comment. Here, "downslope forces generated by gravitational water distribution" refers to the component of the slope weight acting along the slope surface, which contributes to driving forces for slope failure. As rainfall infiltrates the soil, the overall weight of the slope increases due to added water content, leading to a larger downslope force component and thus an increased potential for slope instability.

7. l89: What are the "geoscience factor weights"? In any case, multiple techniques for landslide susceptibility exist, and the references at the end of the sentence are insufficient.

**Response:**

Thank you for your comment. The term "geoscience factor weights" refers to methods that assign weights to geological and environmental factors (e.g., slope, lithology, land use) to quantify their relative contribution to landslide occurrence. Such weighting approaches are commonly based on expert judgment or methods like Analytic Hierarchy Process (AHP). We acknowledge that multiple techniques exist for landslide susceptibility assessment, including statistical models, machine learning methods, and multi-criteria evaluation. To address your concern, we have expanded the references to include representative studies covering various approaches. The revised sentence now reads:

**Introduction (Revised manuscript, line 82)**

"*Substantial efforts have been made to assess landslide susceptibility using various methodologies, including geoscience factor weighting, statistical models, machine learning, and Geographic Information Systems (GIS)-based spatial analysis (Abay et al., 2019; Gebrehiwot et al., 2025; Guo et al., 2023; Pham et al., 2018; Sun et al., 2024; Wang et al., 2024).*"

**References**

Abay, A., Barbieri, G., & Woldearegay, K. (2019). GIS-based Landslide Susceptibility Evaluation Using Analytical Hierarchy Process (AHP) Approach: The Case of Tarmaber District, Ethiopia. *Momona Ethiopian Journal of Science*, 11(1), 14–36. https://doi.org/10.4314/mejs.v11i1

Gebrehiwot, A., Berhane, G., Kide, Y. *et al.* Landslide susceptibility mapping in Lesalso (Laelay Maichew), Northern Ethiopia: a GIS approach using frequency ratio and analytical hierarchy process methods. *Model. Earth Syst. Environ.* **11**, 421 (2025). https://doi.org/10.1007/s40808-025-02578-7

Guo, Z., Guo, F., Zhang, Y., He, J., Li, G., Yang, Y., & Zhang, X. (2023). A python system for regional landslide susceptibility assessment by integrating machine learning models and its application. *Heliyon*, 9(11) https://doi.org/10.1016/j.heliyon.2023.e21542

Pham, B. T., Prakash, I., & Tien Bui, D. (2018). Spatial prediction of landslides using a hybrid machine learning approach based on Random Subspace and Classification and Regression Trees. *Geomorphology*, 303, 256-270. https://doi.org/10.1016/j.geomorph.2017.12.008

Sun, D., Wang, J., Wen, H., Ding, Y., & Mi, C. (2024). Landslide susceptibility mapping (LSM) based on different boosting and hyperparameter optimization algorithms: A case of Wanzhou District, China. *Journal of Rock Mechanics and Geotechnical Engineering*, 16(8), 3221-3232. https://doi.org/10.1016/j.jrmge.2023.09.037

Wang, Y., Ling, Y., Chan, T. O., & Awange, J. (2024). High-resolution earthquake-induced landslide hazard assessment in Southwest China through frequency ratio analysis and LightGBM. *International Journal of Applied Earth Observation and Geoinformation*, 131, 103947. https://doi.org/10.1016/j.jag.2024.103947

8. l93: "few studies" → please cite them.

**Response:**

Thank you for your comment. Due to substantial revisions of the Introduction to improve clarity and structure, the original sentence has been modified. It now reads: "*Many studies provide qualitative descriptions of factor influences but lack quantitative analysis of spatial correlations and interactive effects among multiple driving factors (Shu et al., 2025; Triplett et al., 2025).*" Corresponding references have been added to support this statement.

**References**

Shu, H., Qi, S., Liu, X., Shao, X., Wang, X., Sun, D., ... & He, J. (2025). Relationship between continuous or discontinuous of controlling factors and landslide susceptibility in the high-cold mountainous areas, China. *Ecological Indicators*, *172*, 113313. https://doi.org/10.1016/j.ecolind.2025.113313

Triplett, L.D., Hammer, M.N., DeLong, S.B. *et al.* Factors influencing landslide occurrence in low-relief formerly glaciated landscapes: landslide inventory and susceptibility analysis in Minnesota, USA. *Nat Hazards* **121**, 11799–11827 (2025). https://doi.org/10.1007/s11069-025-07262-8

9. l110: "frequent" again — misleading: it suggests landslides mostly occur in vegetated areas.

**Response:** Thank you for your comment. The original sentence containing "frequent" has been replaced with another expression in the revised Introduction. In addition, all other similar instances in the manuscript have been revised to use "may occur" to accurately reflect the possibility of landslides in areas with high vegetation cover

10. l113: GeoDetector and structural equation modeling require references.

**Response:** Thank you for your comment. To improve the clarity, scientific rigor, structure, and overall quality of the Introduction, we have substantially revised this section. In the revised Introduction, the original sentence has been removed, but references supporting the application of GeoDetector and structural equation

modeling (SEM) have been added to the Research Methods section to ensure the rigor of the methodology. *(e.g.,Lu et al., 2024; Fan et al., 2016; Wang et al., 2010)."*

**References**

Lu, F., Zhang, G., Wang, T., Ye, Y., Zhen, J., & Tu, W. (2024). Analyzing spatial non-stationarity effects of driving factors on landslides: a multiscale geographically weighted regression approach based on slope units. Bulletin of Engineering Geology and the Environment, 83(10), 394. http://doi.org/10.1007/s10064-024-03879-4

Fan, Y., Chen, J., Shirkey, G., John, R., Wu, S. R., Park, H., & Shao, C. (2016). Applications of structural equation modeling (SEM) in ecological studies: an updated review. Ecological Processes, 5(1), 19. http://doi.org/10.1186/s13717-016-0063-3

Wang, J. F., Li, X. H., Christakos, G., Liao, Y. L., Zhang, T., Gu, X., & Zheng, X. Y. (2010). Geographical Detectors-Based Health Risk Assessment and its Application in the Neural Tube Defects Study of the Heshun Region, China. International Journal of Geographical Information Science, 24(1), 107-127. http://doi.org/10.1080/13658810802443457

11. l127: Forest is 65%. Which type of forest? And what about the remaining 35%?

**Response:**

Thank you for your comment. The 65.55% forest coverage refers to the overall forested area in the study region. The remaining area consists of non-forested land, including cropland, grassland, and other land uses. We made minor modifications to line 138 of the original manuscript.

**2.1 Study area (Revised manuscript line 133)**

*"Forest covers 65.55% of the area, while the remaining land consists of non-forested terrain. Vegetation is classified into shrubland, meadow, broadleaf forest, coniferous forest, and cultivated plants. Shrubland is dominated by Myrica and Rhododendron, broadleaf forests include Arundinaria-dominated forests, Quercus*

*engleriana forests, and Castanopsis forests, and coniferous forests consist mainly of Abies forests, Pinus yunnanensis forests, and subalpine Quercus forests.*"

12. l127: "terrain slopes from southwest to northeast" — what do you mean? This sounds strange.

**Response:**

Thank you for pointing this out. The original phrase "terrain slopes from southwest to northeast" was intended to indicate the overall elevation trend in the study area, with lower elevations in the southwest and higher elevations in the northeast. To avoid potential misunderstanding, we have removed this phrase. The elevation range and vertical drop are retained, providing clear information about topography: elevations range from 530 to 3,321 m, with a vertical drop of 2,700 m.

13. l137: It is unclear why you introduce this major landslide and how it connects to the large-scale analysis. This point is crucial.

**Response:**

Thank you very much for your insightful comment. We agree that the connection between the described major landslide and the large-scale susceptibility analysis needs to be clarified. The inclusion of this landslide aims to serve as a representative case study that bridges our regional-scale susceptibility assessment with the mechanistic understanding at the site scale. Specifically, this event occurred within an area classified as a moderate-susceptibility *zone* in our AHP-based regional evaluation. Although it was not located in a high-susceptibility area, the landslide was triggered by the combined effects of prolonged rainfall, anthropogenic loading from waste deposition, and the additional weight of dense vegetation. These local conditions significantly increased slope instability, leading to failure even in a zone that was only moderately susceptible at the regional scale. This observation highlights the amplifying influence of vegetation and rainfall interactions, which supports the "double-edged sword" role of vegetation revealed by our watershed-scale GeoDetector and SEM analyses. By presenting this case, we aim to demonstrate how

local environmental disturbances can alter slope stability beyond regional susceptibility levels, thus linking large-scale statistical results to field-scale mechanisms. Manuscript content modifications regarding the major landslide and its relation to the large-scale analysis have been provided in Comment 1.

14. l174: SoilGrids (Hengl et al., 2017) describes soils, not lithology. How did you derive lithology from those data?

**Response:**

Thank you for pointing this out. The citation was incorrect in the previous version. The lithology data used in this study were actually derived from the Global Lithological Map (GLiM), which provides global-scale information on rock types and lithological properties. The correct reference is as follows:

" *Hartmann, J., & Moosdorf, N. (2012). The new global lithological map database GLiM: A representation of rock properties at the Earth surface. Geochemistry, Geophysics, Geosystems, 13(12), Q12004. https://doi.org/10.1029/2012GC004370*"

15. l178: Why "maximum"? Please clarify.

**Response:**

Thank you for your comment. The maximum NDVI raster was used to represent the densest or most developed vegetation in each pixel. In our study area, which generally exhibits high vegetation coverage, the maximum NDVI effectively captures the characteristic high vegetation conditions, while reducing the influence of temporary vegetation loss or seasonal variations. This provides a consistent and representative indicator of vegetation cover for landslide susceptibility modeling. It is worth noting that the maximum NDVI usually occurs in summer, when vegetation reaches its peak biomass and self-weight. This condition corresponds to the period when vegetation exerts the greatest mechanical influence on slopes, providing a solid basis for investigating its role in enhancing or reducing slope stability in well-vegetated areas.

16. l189: "Landslide hazard point data were obtained from the GeoCloud platform..."
— this sentence is poor. The GeoCloud data are never mentioned again. No description is provided for the method or imagery used to prepare the inventory. Were these recent landslides? Is it a geomorphological inventory? What imagery and what dates were used? Without this, the inventory's quality cannot be assessed — a major issue.

**Response:**

Thank you for your valuable comment. In this study, the landslide inventory was compiled from two sources. The first source was the officially released landslide inventory of the Jinkouhe area provided by the GeoCloud platform, which mainly includes historical landslides with detailed attribute information such as geographic coordinates, township location, landslide scale, hazard level, occurrence time, and associated losses. The second source consisted of landslides identified through visual interpretation of high-resolution Sentinel-2 imagery acquired in August 2024 and subsequent manual verification. The datasets from both sources were integrated, and duplicate or uncertain cases were removed, resulting in a total of 227 landslides, representing the complete landslide distribution within the study area. We have revised the manuscript to provide a clearer and more detailed description of the landslide inventory and its preparation process. To address the unclear description in the original manuscript, this has been revised in # Revised manuscript line 196. Specific revisions related to this issue are detailed in our response to Comment 3.

17. l196: "Landslide susceptibility analysis was conducted by overlaying landslide sites with…" → You only overlaid the landslide points?

**Response:**

Thank you for your comment. We believe this issue may stem from a misunderstanding of the methodological description. The phrase "overlaying landslide sites with conditioning factors" does not simply refer to a basic overlay of landslide point data, but rather to a multi-factor spatial analysis conducted within the framework

of the Analytic Hierarchy Process (AHP). Specifically, each conditioning factor (elevation, slope, aspect, distance to faults, rivers, and roads, lithology, rainfall, NDVI, and wind speed) was classified, and buffer and statistical analyses were performed to determine the number of landslides occurring within each category. These results served as one of the important bases for determining the weights of the factors. Subsequently, all factor layers were weighted and overlaid according to the AHP-derived weights, and consistency tests were performed to ensure reliability. Finally, a landslide susceptibility distribution map was generated. The detailed procedure has been described in the main text, and the classification criteria and statistical results for each factor are provided in Appendix Text S1. To avoid ambiguity, we have revised the corresponding description in the revised manuscript at line 206.

**2.3.1 landslide susceptibility based on the analytic hierarchy process(Revised manuscript line 207)**

"*Following the best-practice guidelines for landslide susceptibility mapping proposed by Reichenbach et al. (2018), landslide susceptibility analysis was conducted using a multi-factor spatial evaluation approach within the framework of the AHP. The analysis considered the following conditioning factors: elevation, slope, aspect, distances to faults, rivers, and roads, lithology, rainfall, NDVI, and wind speed. All factors passed the multicollinearity test with variance inflation factor (VIF) values below 10 (Arabameri et al., 2019; Chen et al., 2018). Range normalization was applied to standardize all indicators (He et al., 2024). Each factor was then classified, and to quantify its influence, buffer and statistical analyses were performed to calculate the number of landslides occurring within different classified zones, serving as one of the key bases for determining the factor weights (Table S1). The factors were subsequently weighted through AHP and validated using consistency tests. Finally, the overlay analysis produced a landslide susceptibility distribution map of the study area (Ahmad et al., 2023; Asmare, 2023), with susceptibility categorized into five levels: very low, low, medium, high, and very high, based on existing standards.*"

**References**

Ahmad, M. S., MonaLisa, & Khan, S. (2023). Comparative analysis of analytical hierarchy process (AHP) and frequency ratio (FR) models for landslide susceptibility mapping in Reshun, NW Pakistan. *Kuwait Journal of Science*, 50(3), 387-398. https://doi.org/10.1016/j.kjs.2023.01.004

Arabameri, A., Pradhan, B., Rezaei, K., Sohrabi, M., & Kalantari, Z. (2019). GIS-based landslide susceptibility mapping using numerical risk factor bivariate model and its ensemble with linear multivariate regression and boosted regression tree algorithms. *Journal of Mountain Science*, 16(3), 595-618. http://doi.org/10.1007/s11629-018-5168-y

Asmare, D. (2023). Application and validation of AHP and FR methods for landslide susceptibility mapping around choke mountain, northwestern ethiopia. *Scientific African*, 19, e1470. https://doi.org/10.1016/j.sciaf.2022.e01470

Chen, W., Li, H., Hou, E., Wang, S., Wang, G., Panahi, M., Li, T., Peng, T., Guo, C., Niu, C., Xiao, L., Wang, J., Xie, X., & Ahmad, B. B. (2018). GIS-based groundwater potential analysis using novel ensemble weights-of-evidence with logistic regression and functional tree models. *Science of the Total Environment*, 634, 853-867. https://doi.org/10.1016/j.scitotenv.2018.04.055

He, S., Yang, H., Chen, X., Wang, D., Lin, Y., Pei, Z., Li, Y., & Akbar Jamali, A. (2024). Ecosystem sensitivity and landscape vulnerability of debris flow waste-shoal land under development and utilization changes. Ecological Indicators, 158, 111335. https://doi.org/10.1016/j.ecolind.2023.111335

Reichenbach, P., Rossi, M., Malamud, B. D., Mihir, M., & Guzzetti, F. (2018). A review of statistically-based landslide susceptibility models. Earth-science reviews, 180, 60-91. https://doi.org/10.1016/j.earscirev.2018.03.001

18. l238: "An equal number of non-landslide points were created" — how?

**Response:** Thank you for your insightful comment. To assess the reliability of the dataset and the robustness of the model, a 500 m buffer was generated around the

known landslide points, and areas close to water bodies were excluded. Within the remaining regions, non-landslide points (negative samples) were randomly selected while maintaining a balanced ratio between positive and negative samples.

**2.3.2 Rationality validation of susceptibility assessment result(Revised manuscript line 252)**

"*Based on the susceptibility distribution map and known landslide points, non-landslide points were randomly sampled from areas excluding water bodies and 500 m landslide buffer zones to maintain a balanced ratio between positive and negative samples. Following the repeated random sampling approach commonly used in machine learning studies, 30% of the entire dataset was repeatedly and randomly selected as the testing subset for ROC/AUC evaluation to assess model robustness.*"

19. l253: The GeoDetector method should be applied with a large number of landslides; are 227 sufficient?

**Response:**

Thank you for raising this important question. According to previous studies, the Geodetector method requires an adequate sample size to ensure the reliability of variance decomposition. However, GeoDetector does not rely on a large number of samples; instead, it emphasizes the spatial consistency between the dependent variable and explanatory factors (Wang et al., 2010). Based on the recommendations of Wang and Xu (2017) and subsequent research (Zhou et al., 2021), a sample size ranging from 100 to 500 points is generally sufficient to ensure stable q-statistic estimation. In this study, a total of 227 landslide samples were used, which falls within the recommended range and meets the statistical assumptions of the Geodetector. In addition, the grid-based sampling design ensured a sufficient number of spatial strata for factor detection, thereby enhancing the robustness of the analysis. Therefore, we consider that the sample size is adequate for applying the GeoDetector approach in this study.

**References**

Wang, J., Li, X., Christakos, G., Liao, Y., Zhang, T., Gu, X., & Zheng, X. (2010).
Geographical Detectors-Based Health Risk Assessment and its Application in the
Neural Tube Defects Study of the Heshun Region, China. *International Journal
of Geographical Information Science*, 24(1), 107–127.
https://doi.org/10.1080/13658810802443457

Wang, J., & Xu, C. (2017). Geodetector: Principle and prospect. *Acta Geographica
Sinica*, 72(1), 116–134. https://doi.org/10.11821/dlxb201701010

Zhou, X., Wen, H., Zhang, Y., Xu, J., & Zhang, W. (2021). Landslide susceptibility
mapping using hybrid random forest with GeoDetector and RFE for factor
optimization. *Geoscience Frontiers*, 12(5), 101211.
https://doi.org/10.1016/j.gsf.2021.101211

20. l258: Why did you use landslide susceptibility as the dependent variable? It is already a model. Why not use the landslide inventory?

**Response:**

Thank you for your insightful comment. In conventional GeoDetector applications, the landslide inventory is usually used as the dependent variable to identify the main conditioning factors influencing landslide occurrence. However, treating landslide and non-landslide samples as a binary (0–1) dependent variable is more suitable for factor screening and cannot effectively explain how conditioning factors interact to influence landslide susceptibility (Liu et al., 2024; Sun et al., 2021; Yang et al., 2019; Zhou et al., 2023;). Consequently, most previous studies using this approach have focused on identifying dominant factors rather than revealing the underlying mechanisms. In contrast, the present study employed the GeoDetector for a different purpose—to quantify the explanatory power of each conditioning factor on the *modeled landslide susceptibility* derived from the AHP framework and, more importantly, to reveal how factor interactions contribute to susceptibility patterns. This design enhances the interpretability and physical consistency of the susceptibility model (Chen et al., 2023).

**References**

Chen, Z., Song, D. & Dong, L. An innovative method for landslide susceptibility mapping supported by fractal theory, GeoDetector, and random forest: a case study in Sichuan Province, SW China. Nat Hazards 118, 2543–2568 (2023). https://doi.org/10.1007/s11069-023-06104-9

Liu, X., Shao, S., & Shao, S. (2024). Landslide susceptibility prediction and mapping in Loess Plateau based on different machine learning algorithms by hybrid factors screening: Case study of Xunyi County, Shaanxi Province, China. Advances in Space Research, 74(1), 192-210. https://doi.org/10.1016/j.asr.2024.03.074

Sun, D., Shi, S., Wen, H., Xu, J., Zhou, X., & Wu, J. (2021). A hybrid optimization method of factor screening predicated on GeoDetector and Random Forest for Landslide Susceptibility Mapping. Geomorphology, 379, 107623. https://doi.org/10.1016/j.geomorph.2021.107623

Yang, J., Song, C., Yang, Y., Xu, C., Guo, F., & Xie, L. (2019). New method for landslide susceptibility mapping supported by spatial logistic regression and GeoDetector: A case study of Duwen Highway Basin, Sichuan Province, China. Geomorphology, 324, 62-71.https://doi.org/10.1016/j.geomorph.2018.09.019

Zhou, X., Wen, H., Zhang, Y., Xu, J., & Zhang, W. (2021). Landslide susceptibility mapping using hybrid random forest with GeoDetector and RFE for factor optimization. Geoscience Frontiers, 12(5), 101211. https://doi.org/10.1016/j.gsf.2021.101211

21. l259: what's "geological hazard risk" — a scientific paper should use terms accurately.

**Response:**

Thank you for your comment. We agree that the term should be more precise. The term "geological hazard risk" in this part has been revised to "*landslide hazard risk*" to ensure terminology accuracy.

22. l324: "84 landslides covering 53.77% of the total area" — you said landslides are points: how can they cover an area? In pixels?

**Response:**

Thank you for your comment. We acknowledge the potential confusion. The percentages reported (e.g., 53.77% of the total area) refer to the proportion of the study area classified as moderate landslide susceptibility, not the area physically covered by the landslide points themselves. The numbers of landslides indicate how many inventory points fall within each susceptibility class, while the percentages describe the corresponding area of each class. We have revised the text to clarify this distinction.

**3.1 Landslide susceptibility mapping and distribution characteristics(Revised manuscript line 364)**

"*Most of the study area shows moderate landslide susceptibility, occupying 53.77% of the total area and containing 84 landslide points.*"

23. l336: This section pops up quite suddenly and without any clear justification, in my opinion.

**Response:**

Thank you for this comment. We agree that the purpose of this section was not sufficiently clarified in the original manuscript. The H/L ratio is introduced to provide a quantitative characterization of landslide mobility and runout behavior, which helps place the representative landslide case in the context of landslide magnitude and potential hazard. In the revised manuscript, we have added a brief justification at the beginning of Section 3.2 to clarify its relevance and to explain how this analysis supports the interpretation of landslide dynamics discussed later in the paper.

**3.2 Relationship between H/L and area of a typical landslide(Revised manuscript line 378)**

*"To quantitatively characterize the mobility and runout behavior of the representative landslide, the relationship between landslide height (H), travel distance (L), and affected area was analyzed."*

24. l400: "faults occur where the structural stability of the slopes is poor" — this is scientifically incorrect; the causal direction is reversed.

**Response:**

Thank you for your careful review. We agree with your comment. The original sentence reversing the causal relationship has been deleted. The revised text now reads:

**# Revised manuscript line 442**

"*This is because faults zones can cause localized stress and weaken the structural integrity of slopes, especially in the study area, which lies at the intersection of the Longquanshan Fault Zone and the Ebian-Mabian Seismic Belt located in the central segment of China's north–south seismic belt.*"

25. l462: "public factors" — what do you mean by this?

**Response:** Thank you for pointing out this inconsistency. This was an oversight on our part. The term "public factors" on line 462 was used in error. Our intended term, as defined in the Methods section (Line 231), is "common factors", which refers to [elevation, slope, aspect, lithology, and distances to faults, rivers, and roads.] the factors that are shared across different factor combinations in our analysis. We have corrected "public factors" to "common factors" on line 504 in the revised manuscript to maintain terminological consistency throughout the paper.

26. l500: The statements throughout this section are not supported by data analysis, in my opinion.

**Response:**

We thank you for highlighting the need for stronger quantitative support in Section

4.3 regarding the "ambiguous role" of vegetation. We agree that some statements in this subsection were more speculative and have now revised this section 4.3 better integrate our empirical findings and quantitative results. Specifically, we have:

1.Explicitly linked the discussion to our quantitative results from GeoDetector and SEM (e.g., NDVI's interaction effects, total effect coefficients), as well as slope stability calculations under saturated vs. natural conditions.

2.Replaced speculative statements with evidence-based interpretations, using data from our susceptibility scenarios (Categories I–V) and stability factor (Fs) values to explain how vegetation's role shifts with rainfall and slope conditions.

3.Clarified that the "ambiguity" is not merely hypothetical, but is demonstrated through: The bifactor enhancement between NDVI and rainfall (Fig. 8), showing that vegetation can amplify rainfall's impact in certain contexts; The decrease in slope stability (Fs) from 1.13 to 0.89 under saturated conditions when vegetation weight is considered (Table 3), providing direct mechanical evidence of its potential destabilizing effect; The shifts in susceptibility zoning when vegetation is added to the model (Table 6), illustrating its spatially varying influence.

We believe these revisions strengthen the subsection by grounding the discussion in our own analytical results, thereby providing a more substantiated explanation of vegetation's dual role.

**4.3 Mechanisms of landslides in areas with high vegetation coverage (Revised manuscript line 543)**

"*The mechanisms underlying landslide initiation in densely vegetated areas are complex and context-dependent, as evidenced by the contrasting effects of vegetation revealed in our multi-scale analysis. Our findings demonstrate that vegetation does not act uniformly as a stabilizer; rather, its role is modulated by hydrological conditions, slope gradient, and external loading.*

*At the watershed scale, the GeoDetector results indicate that NDVI alone exhibits limited independent explanatory power (q = 0.27, Table 4). However, its interaction with rainfall significantly enhances landslide susceptibility (e.g., NDVI × rainfall q =*

0.67, Fig. 8), suggesting that vegetation can amplify the destabilizing effects of precipitation under certain conditions. While vegetation intercepts rainfall and promotes evapotranspiration, it can also alter soil moisture distribution via stemflow, root-induced preferential flow, and reduced surface runoff. Under prolonged rainfall, these processes may lead to localized saturation, thereby exacerbating landslide and debris flow risks in vegetated slopes. This aligns with the SEM results, which attribute a total indirect effect of 0.21 to NDVI, mediated largely through soil moisture dynamics and interactions with rainfall and slope (Fig. 9, Table 5). The susceptibility scenario analysis further illustrates this duality: adding vegetation alone (Class II) slightly reduced the extent of very high susceptibility zones, yet when combined with rainfall (Class IV) and wind (Class V), it led to a notable expansion of high-susceptibility areas and an increase in landslide counts (Table 6, Fig. 10). This suggests that vegetation's protective capacity may be offset or reversed under prolonged rainfall, especially on steeper slopes.

At the site-specific scale, the stability calculations provide direct mechanical insight into how vegetation can transition from a stabilizing to a destabilizing factor. Under natural (unsaturated) conditions, the slope remained stable even with the added weight of vegetation and waste material (Fs = 1.02). However, under saturated conditions, the same additional loads—particularly the self-weight of trees—reduced the stability coefficient to 0.89, triggering failure (Table 3). This demonstrates that the mechanical reinforcement from roots can be outweighed by the gravitational load of vegetation when soil strength is reduced by saturation, a shift that is quantitatively captured by our modeling.

These findings help explain why landslides may occur unexpectedly in densely vegetated areas. Vegetation can create a false sense of stability by masking early signs of movement (e.g., surface cracking, minor slumping) and by being traditionally associated with slope protection. Moreover, the same root networks that enhance soil cohesion also facilitate preferential infiltration, potentially accelerating soil saturation during heavy rainfall—a process reflected in the strong interaction between NDVI and rainfall in our spatial analysis. In terrain with high lateral variability in slope, lithology, or soil depth, vegetation may thus contribute to highly localized and concealed instability, as exemplified by the 2023 Jinkouhe landslide.

In summary, our integrated analysis provides quantitative evidence that

*vegetation's role is not merely "ambiguous" in a speculative sense, but is quantifiably dual: it stabilizes slopes through root reinforcement under moderate conditions, yet can promote instability through added weight, enhanced infiltration, and synergistic interactions with rainfall when critical thresholds are exceeded. This duality underscores the importance of considering vegetation not as a static stabilizing factor, but as a dynamic component of the hillslope system in landslide susceptibility assessments.*"

We sincerely appreciate your constructive feedback. We hope the revisions and responses provided will ensure our manuscript meets the standards for publication in ***Natural Hazards and Earth System Sciences***.

---

## Author Comment (AC3)

**Response letter**

**Dear Editors and Reviewers:**

**Re: egusphere-2025-3004**

We sincerely thank you and the reviewers for providing us with such a valuable revision opportunity. Thus, we can further improve and present our studies. The comments from you and the reviewers were highly insightful and enabled us to greatly improve the quality of our manuscript. We have carefully reviewed the feedback and made corrections that we hope will be met with approval. Revised portions are marked on the revised manuscript. Please note that these resulting revisions did not change the paper's findings.

In the response letter to the editor and reviewers, we firstly summarized the major changes in a cover letter to the editors, and we then itemized our response to editors and reviewers, **in which the blue font indicates the response to each comment and the black font presents the revision from the revised manuscript.**

We hope that the revisions in the revised manuscript and the responses to the comments will suffice to allow our manuscript to be suitable for publication in ***Natural Hazards and Earth System Sciences***.

**Sincerely regards,**

**Songtang He (hest@imde.ac.cn)**

**Institute of Mountain Hazards and Environment, Chinese Academy of Sciences**

**Response to Reviewer #3**

**[Comment 1] The manuscript focuses on understanding and evaluating the key issue of vegetation's mitigating effect on rainfall-induced landslides. However, I confirm what RC1 already observed, that the way in which vegetation is considered is also the main critical point of the study. In fact, although the aim of the study is to better understand its effect on triggering landslides, it has been treated too superficially. Furthermore, the results show an average landslide height of 36 m, so it is not surprising to see a total effect coefficient value of 0.21. For this type of deep landslide, the vegetation's effect is definitely limited. I agree with the comments already made by RC1, so I will try not to be repetitive based on what has already been highlighted.**

**Response:**

We greatly appreciate your constructive perspective on the relationship between landslide depth and vegetation effects, and we recognize the value of these insights for enhancing the scientific rigor of our study. We will carefully revise the manuscript to address your concerns and strengthen the overall quality of the work. According to the comments from RC1, we have fully revised the manuscript. You can also check the response to the RC1. Here, it should point out 36 m is not the average height, but the width. Meanwhile, the total effect coefficient value of vegetation (0.21) is relatively small. But in terms of the top ten impact factor rankings, it still holds a leading position. That's why we want to emphasize the role of vegetation on landslide susceptibility.

**[Comment 2] The introduction needs significant improvement. It is currently written in a confusing form and lacks a clear structure that would help the reader to follow the study. In my opinion, it would be useful to provide more information about the available literature, rather than just citing studies and quoting short sentences. Also, it is necessary to clarify the research gap by explaining its importance and how this study contributes to bridging it.**

**Response:**

Thank you for this valuable comment. We agree that the previous version of the Introduction required substantial improvement in terms of structure, clarity, and articulation of the research gap.

Accordingly, the Introduction has been thoroughly revised and reorganized to enhance its logical flow and readability. The revised text now provides a more coherent narrative, beginning with the broader context of landslide hazards in vegetated mountainous regions, followed by an integrated discussion of the dual role of vegetation in slope stability. Rather than simply listing prior studies, we expanded the description of existing literature to synthesize current understanding of both the stabilizing and destabilizing effects of vegetation, particularly in relation to rainfall-driven landslide processes.

In addition, the research gap is now clarified in a more explicit and focused manner. The revised introduction emphasizes the limitations of existing studies in fully capturing the complex interactions among vegetation and other environmental controls on landslide susceptibility, as well as the need for approaches that link regional susceptibility patterns with site-specific failure mechanisms. On this basis, the contribution of the present study is clearly positioned as a multi-scale investigation that integrates statistical analysis and geomechanical modeling to better explain vegetation–landslide interactions.

We believe that these revisions significantly improve the clarity, structure, and scientific focus of the Introduction, and more clearly demonstrate the relevance and contribution of this study. We appreciate your insightful suggestions, which have helped strengthen the manuscript. Please check it below and in manuscript in lines 54-116, pages 4-6.

**1 Introduction**

"*Landslides represent a significant geological hazard in mountainous regions worldwide, causing substantial loss of life, infrastructure damage, and economic disruption (Alvioli et al., 2024; Zhang et al., 2025). In areas with dense vegetation cover, the relationship between vegetation and slope stability is particularly complex*

and non-linear (Deng et al., 2022; Medina et al., 2021). While vegetation is traditionally regarded as a stabilizing agent through root reinforcement, soil moisture regulation, and erosion control (He et al., 2017; Lan et al., 2020; Rey et al., 2019), shallow landslides may occur even in densely vegetated landscapes (Xu et al., 2024). This paradox underscores the dual—and often contradictory—role of vegetation in landslide processes, acting as both a mitigating and a predisposing factor depending on environmental context and trigger conditions.

The stabilizing function of vegetation is well-documented. Root systems enhance soil cohesion and shear strength, while canopy and litter layers reduce rainfall impact and surface runoff (Gonzalez-Ollauri & Mickovski, 2016; Murgia et al., 2022; Vergani et al., 2017). However, under certain conditions, vegetation can exacerbate slope instability. The added weight of trees, especially on steep slopes, increases gravitational driving forces (Schmaltz & Mergili, 2018). Vegetation can also alter soil hydrological properties, increasing infiltration and soil moisture content, which in turn reduces effective stress and shear resistance during rainfall events (Qin et al., 2022). Furthermore, wind forces acting on tall vegetation can transmit dynamic loads to the slope, while root wedging in thin soils may promote fracture development (Bordoloi & Ng, 2020; Liu et al., 2020). Rainfall remains the primary trigger of landslides in vegetated areas, as it saturates the soil, elevates pore water pressure, and reduces slope stability (Dhanai et al., 2022; Li et al., 2025). Therefore, landslide initiation in vegetated terrain is not governed by vegetation alone but results from the intricate interplay among vegetation characteristics, rainfall intensity, slope gradient, lithology, and other environmental factors.

Substantial efforts have been made to assess landslide susceptibility using various methodologies, including geoscience factor weighting, statistical models, machine learning, and Geographic Information Systems (GIS)-based spatial analysis (Abay et al., 2019; Gebrehiwot et al., 2025; Guo et al., 2023; Pham et al., 2018; Sun et al., 2024; Wang et al., 2024). These approaches have improved our understanding of the spatial distribution of landslides and the relative importance of conditioning factors. However, several critical gaps remain. First, many studies provide qualitative descriptions of factor influences but lack quantitative analysis of spatial correlations and interactive effects among multiple driving factors (Shu et al., 2025; Triplett et al., 2025). Second, while rainfall-landslide relationships have been extensively studied

*using spatial autocorrelation and clustering techniques (Chen et al., 2024; Liu et al., 2024; Ortiz-Giraldo et al., 2023; Pokharel et al., 2021; Wang et al., 2020), the moderating role of vegetation in these relationships is poorly quantified. Specifically, how vegetation mediates the effects of rainfall, lithology, slope, and wind on slope stability coefficients remains unclear (Lan et al., 2020). Third, most susceptibility models operate at a single spatial scale, either regional/watershed or site-specific, with limited integration across scales. This hampers a holistic understanding of how macro-scale predisposing factors translate into micro-scale failure mechanisms.*

*To address these research gaps, this study investigates the dual-edged role of vegetation in landslide susceptibility by integrating watershed-scale statistical analysis with site-specific geomechanical modeling. We selected the Jinkouhe District in Southwest China⸺a region with high vegetation cover (≥65.5%) and frequent landslide activity⸺as our study area. The research aims to (1) Quantify the individual and interactive effects of key environmental factors (rainfall, vegetation, wind speed, slope, lithology, etc.) on landslide susceptibility at the watershed scale using Geodetector and Structural Equation Modeling (SEM). (2) Analyze the mechanical role of vegetation weight and its coupling with rainfall and anthropogenic loading in triggering a typical shallow landslide through slope stability calculations. (3) Integrate findings from both scales to elucidate how vegetation mediates landslide processes under different environmental conditions, thereby providing a multi-scale perspective on its ＂double-edged sword＂ function. By bridging macroscopic susceptibility patterns with microscopic failure mechanisms, this study offers novel insights into the complex vegetation‒landslide interplay. The results are expected to enhance the accuracy of landslide risk assessments and inform sustainable slope management strategies in densely vegetated mountainous regions."*

**References:**

Abay, A., Barbieri, G., & Woldearegay, K. (2019). GIS-based Landslide Susceptibility Evaluation Using Analytical Hierarchy Process (AHP) Approach: The Case of Tarmaber District, Ethiopia. Momona Ethiopian Journal of Science, 11(1), 14–36. https://doi.org/10.4314/mejs.v11i1.

Alvioli, M., Loche, M., Jacobs, L., Grohmann, C. H., Abraham, M. T., Gupta, K., Satyam, N., Scaringi, G., Bornaetxea, T., Rossi, M., Marchesini, I., Lombardo, L.,

Moreno, M., Steger, S., Camera, C. A. S., Bajni, G., Samodra, G., Wahyudi, E. E., Susyanto, N., Sinčić, M., Gazibara, S. B., Sirbu, F., Torizin, J., Schüßler, N., Mirus, B. B., Woodard, J. B., Aguilera, H., & Rivera-Rivera, J. (2024). A benchmark dataset and workflow for landslide susceptibility zonation. *Earth-Science Reviews*, 258, 104927. https://doi.org/10.1016/j.earscirev.2024.104927

Bordoloi, S., & Ng, C. W. W. (2020). The effects of vegetation traits and their stability functions in bio-engineered slopes: A perspective review. *Engineering Geology*, 275, 105742. https://doi.org/10.1016/j.enggeo.2020.105742

Chen, C., Liu, Y., Li, Y., & Guo, F. (2024). Mapping landslide susceptibility with the consideration of spatial heterogeneity and factor optimization. *Natural Hazards* http://doi.org/10.1007/s11069-024-06955-w

Deng, J., Ma, C., & Zhang, Y. (2022). Shallow landslide characteristics and its response to vegetation by example of July 2013, extreme rainstorm, Central Loess Plateau, China. *Bulletin of Engineering Geology and the Environment*, 81(3), 100. http://doi.org/10.1007/s10064-022-02606-1

Dhanai, P., Singh, V.P. & Soni, P. Rainfall Triggered Slope Instability Analysis with Changing Climate. *Indian Geotech J* **52**, 477–492 (2022). https://doi.org/10.1007/s40098-021-00581-0

Gebrehiwot, A., Berhane, G., Kide, Y. *et al.* Landslide susceptibility mapping in Lesalso (Laelay Maichew), Northern Ethiopia: a GIS approach using frequency ratio and analytical hierarchy process methods. *Model. Earth Syst. Environ.* **11**, 421 (2025). https://doi.org/10.1007/s40808-025-02578-7

Gonzalez-Ollauri, A., & Mickovski, S. B. (2016). Using the root spread information of pioneer plants to quantify their mitigation potential against shallow landslides and erosion in temperate humid climates. *Ecological Engineering*, 95, 302-315. https://doi.org/10.1016/j.ecoleng.2016.06.028

Guo, Z., Guo, F., Zhang, Y., He, J., Li, G., Yang, Y., & Zhang, X. (2023). A python system for regional landslide susceptibility assessment by integrating machine learning models and its application. *Heliyon*, 9(11) https://doi.org/10.1016/j.heliyon.2023.e21542

He, S., Wang, D., Fang, Y., & Lan, H. (2017). Guidelines for integrating ecological and biological engineering technologies for control of severe erosion in mountainous areas – A case study of the Xiaojiang River Basin, China. *International Soil and Water Conservation Research*, 5(4), 335-344. https://doi.org/10.1016/j.iswcr.2017.05.001

Lan, H., Wang, D., He, S., Fang, Y., Chen, W., Zhao, P., & Qi, Y. (2020). Experimental study on the effects of tree planting on slope stability. *Landslides*, 17(4), 1021-

1035. http://doi.org/10.1007/s10346-020-01348-z

Li, Z., Guo, J., Li, T. *et al.* Influence of topography on the fragmentation and mobility of landslides. *Bull Eng Geol Environ* **84**, 73 (2025). https://doi.org/10.1007/s10064-025-04095-4

Liu, W., Yang, Z., & He, S. (2020). Modeling the landslide-generated debris flow from formation to propagation and run-out by considering the effect of vegetation. *Landslides*, 18, 43-58. https://doi.org/10.1007/s10346-020-01478-4

Medina, V., Hürlimann, M., Guo, Z., Lloret, A., & Vaunat, J. (2021). Fast physically-based model for rainfall-induced landslide susceptibility assessment at regional scale. *Catena*, 201, 105213. https://doi.org/10.1016/j.catena.2021.105213

Murgia, I., Giadrossich, F., Mao, Z., Cohen, D., Capra, G. F., & Schwarz, M. (2022). Modeling shallow landslides and root reinforcement: A review. *Ecological Engineering*, 181, 106671. https://doi.org/10.1016/j.ecoleng.2022.106671

Ortiz-Giraldo L, Botero BA and Vega J (2023) An integral assessment of landslide dams generated by the occurrence of rainfall-induced landslide and debris flow hazard chain. *Front. Earth Sci.* 11:1157881. http://doi.org/10.3389/feart.2023.1157881

Pham, B. T., Prakash, I., & Tien Bui, D. (2018). Spatial prediction of landslides using a hybrid machine learning approach based on Random Subspace and Classification and Regression Trees. *Geomorphology*, 303, 256-270. https://doi.org/10.1016/j.geomorph.2017.12.008

Pokharel, B., Althuwaynee, O. F., Aydda, A., Kim, S., Lim, S., & Park, H. (2021). Spatial clustering and modelling for landslide susceptibility mapping in the north of the Kathmandu Valley, Nepal. *Landslides*, 18(4), 1403-1419. http://doi.org/10.1007/s10346-020-01558-5

Qin, M., Cui, P., Jiang, Y. et al. Occurrence of shallow landslides triggered by increased hydraulic conductivity due to tree roots. Landslides 19, 2593–2604 (2022). https://doi.org/10.1007/s10346-022-01921-8

Rey, F., Bifulco, C., Bischetti, G. B., Bourrier, F., De Cesare, G., Florineth, F., Graf, F., Marden, M., Mickovski, S. B., Phillips, C., Peklo, K., Poesen, J., Polster, D., Preti, F., Rauch, H. P., Raymond, P., Sangalli, P., Tardio, G., & Stokes, A. (2019). Soil and water bioengineering: Practice and research needs for reconciling natural hazard control and ecological restoration. *Science of the Total Environment*, 648, 1210-1218. http://doi.org/https://doi.org/10.1016/j.scitotenv.2018.08.217

Schmaltz, E. M., & Mergili, M. (2018). Integration of root systems into a GIS-based slip surface model: computational experiments in a generic hillslope environment. *Landslides*, 15(8), 1561-1575. http://doi.org/10.1007/s10346-018-0970-8

Shu, H., Qi, S., Liu, X., Shao, X., Wang, X., Sun, D., ... & He, J. (2025). Relationship between continuous or discontinuous of controlling factors and landslide susceptibility in the high-cold mountainous areas, China. Ecological Indicators, 172, 113313. https://doi.org/10.1016/j.ecolind.2025.113313

Sun, D., Wang, J., Wen, H., Ding, Y., & Mi, C. (2024). Landslide susceptibility mapping (LSM) based on different boosting and hyperparameter optimization algorithms: A case of Wanzhou District, China. *Journal of Rock Mechanics and Geotechnical Engineering*, 16(8), 3221-3232. https://doi.org/10.1016/j.jrmge.2023.09.037

Triplett, L.D., Hammer, M.N., DeLong, S.B. et al. Factors influencing landslide occurrence in low-relief formerly glaciated landscapes: landslide inventory and susceptibility analysis in Minnesota, USA. Nat Hazards 121, 11799–11827 (2025). https://doi.org/10.1007/s11069-025-07262-8

Vergani, C., Giadrossich, F., Buckley, P., Conedera, M., Pividori, M., Salbitano, F., Rauch, H. S., Lovreglio, R., & Schwarz, M. (2017). Root reinforcement dynamics of European coppice woodlands and their effect on shallow landslides: A review. *Earth-Science Reviews*, 167, 88-102. http://doi.org/https://doi.org/10.1016/j.earscirev.2017.02.002

Wang, Y., Feng, L., Li, S., Ren, F., & Du, Q. (2020). A hybrid model considering spatial heterogeneity for landslide susceptibility mapping in Zhejiang Province, China. *Catena*, 188, 104425. http://doi.org/https://doi.org/10.1016/j.catena.2019.104425

Wang, Y., Ling, Y., Chan, T. O., & Awange, J. (2024). High-resolution earthquake-induced landslide hazard assessment in Southwest China through frequency ratio analysis and LightGBM. *International Journal of Applied Earth Observation and Geoinformation*, 131, 103947. https://doi.org/10.1016/j.jag.2024.103947

Xu, Y., Luo, L., Guo, W., Jin, Z., Tian, P., & Wang, W. (2024). Revegetation Changes Main Erosion Type on the Gully–Slope on the Chinese Loess Plateau Under Extreme Rainfall: Reducing Gully Erosion and Promoting Shallow Landslides. *Water Resources Research*, 60(3), e2023WR036307. https://doi.org/10.1029/2023WR036307

Zhang, Y., Li, y., Tom Dijkstra., Janusz Wasowski., Meng, X., Wu, X., Liu, W., Chen, G. (2025). Evolution of large landslides in tectonically active regions - A decade of observations in the Zhouqu County, China. Engineering Geology, 348, 107967. https://doi.org/10.1016/j.enggeo.2025.107967

**[Comment 3] More attention should also be paid to the language used. For example, what is meant by "good vegetation" (do you mean forest density, or plant species, or plant dimensions?), or "higher vegetation cover" (do you**

**mean forest horizontal or vertical structure?), or "vegetation-rich region" (do you mean forest density, or plant species?). All this information is key points in this type of analysis, so I would advise the authors to be more precise and technical.**

**Response:**

We sincerely thank you for your constructive comment regarding the terminology used in our manuscript. We fully acknowledge that the original expressions "good vegetation," "higher vegetation cover," and "vegetation-rich region" could be ambiguous, potentially leading to misinterpretation. In the manuscript, "good vegetation" is intended to refer to areas with high vegetation cover, indicating regions where vegetation is relatively dense; "higher vegetation cover" specifically refers to higher horizontal vegetation cover, emphasizing the horizontal structure of the vegetation rather than vertical structure or height; and "vegetation-rich region" is used to indicate a region with high vegetation density, reflecting areas where plants are densely distributed. In response to your comments, we have systematically revised these terms throughout the manuscript, replacing ambiguous expressions with more explicit and standardized terminology or removing them where appropriate;

**[Comment 4] This section also needs a clear structure_Materials and Methods.**

**Response:**

Thank you for this comment. We agree that a clear and well-organized structure is essential for the Materials and Methods section. Accordingly, the section has been systematically revised and reorganized in the revised manuscript. Following the revisions made in response to the previous comments, the Materials and Methods section is now structured to reflect a clear research logic and multiscale analytical framework.

Specifically, the study is motivated by the observation that landslides and debris flows may still occur in areas with high vegetation density. Therefore, a vegetation-dense region in Southwest China was selected as the study area to investigate the "dual role" of vegetation in landslide processes. Section 2.1 introduces the study area,

including its geographical location, elevation, climate, temperature, and vegetation types, followed by the presentation of a representative landslide case to provide field-based context. Section 2.2 describes the data sources and preprocessing procedures. Section 2.3 presents the modeling strategy for landslide susceptibility assessment and the corresponding accuracy evaluation. Section 2.4 introduces the driving factor analysis methods (GD/SEM) applied to the regional-scale susceptibility results. Finally, Section 2.5 focuses on the physical mechanism analysis at the slope scale, incorporating vegetation self-weight into slope stability analysis.

Through this structure, the revised Materials and Methods section explicitly links regional-scale landslide susceptibility assessment with site-scale slope failure mechanisms, providing a coherent methodological framework that integrates susceptibility mapping with process-based interpretation and offers a new perspective on the role of vegetation in landslide occurrence.

**[Comment 5] There is no information on the morphology or slope of the study area, or on vegetation cover and soil characteristics.**

**Response:**

We thank you for pointing out this omission. In addition to the spatial distribution maps of topography, slope, vegetation cover, and soil characteristics provided in the supporting information (Figure S1), we have also added a more detailed description of vegetation types and their spatial distribution in the study area. Specifically, to provide a more accurate characterization of vegetation, we obtained the vegetation distribution based on the 1:1,000,000 China Vegetation Type Spatial Distribution vector data, as shown in Figure R6. The vegetation in the study area is mainly classified into shrubland, meadow, broadleaf forest, coniferous forest, and cultivated plants. Among them, shrubland mainly consists of Myrica and Rhododendron; broadleaf forests mainly include Arundinaria-dominated forests, Quercus engleriana forests, and Castanopsis forests; and coniferous forests are primarily composed of Abies forests, Pinus yunnanensis forests, and subalpine Quercus forests. This information has been added to Section 2.1 "Study Area" to provide a clearer and more

comprehensive overview of vegetation in the study area.

**2.1 Study Area (Revised manuscript line 126 & 134)**

"*Detailed spatial distributions of topography and vegetation cover are provided in the supporting information (Supplementary Figure S1).*"

"*Vegetation is classified into shrubland, meadow, broadleaf forest, coniferous forest, and cultivated plants. Shrubland is dominated by Myrica and Rhododendron, broadleaf forests include Arundinaria-dominated forests, Quercus engleriana forests, and Castanopsis forests, and coniferous forests consist mainly of Abies forests, Pinus yunnanensis forests, and subalpine Quercus forests.*"

[Figure]

*Fig. R1 Vegetation type distribution map*

**[Comment 6] What are the main plant species, average plant dimensions, the spatial distribution, etc., of your study area?**

**Response:**

We thank you for this comment. The detailed information regarding the main plant species, vegetation types, and their spatial distribution in the study area has already been provided in the previous response and added to Section 2.1 "Study Area" (see

also Figure R1).

**[Comment 7] How are the main soil types distributed in your study area? Furthermore, why was this parameter not considered in the analysis? Main mechanical and hydrological vegetation effects occur in the soil rather than with the lithological substrate.**

**Response:**

We thank you for highlighting the importance of soil characteristics in mediating the mechanical and hydrological effects of vegetation. Indeed, soil plays a key role in slope stability and landslide susceptibility. However, for the Jinkouhe District study area, we were unable to obtain sufficiently high-resolution soil type data (30 m grid), and the available data were of limited reliability, with some soil types covering up to half of the study area. Therefore, lithology was used instead as a proxy in our analysis. Soil types were not included as an independent factor due to these data limitations and to maintain consistency with the factor selection framework applied to other environmental variables. We acknowledge this limitation and plan to incorporate soil type data in future studies when higher-resolution and more reliable data become available.

**[Comment 8] There is insufficient information on the inventory.**

**Response:**

We thank you for pointing out the need for more detailed information regarding the landslide inventory.   In this study, the landslide inventory was compiled from two sources: the first source was the landslide inventory of the Jinkouhe area provided by the GeoCloud platform, and the second source consisted of landslides identified through visual interpretation of high-resolution satellite imagery (Sentinel-2) and manual verification. The datasets were integrated, and duplicate or uncertain cases were removed, resulting in a total of 227 landslides, representing the complete landslide distribution within the study area. To address the unclear description in the original manuscript, this section has been revised in # Revised manuscript line 202. Additionally, to visually demonstrate the reliability of some landslide points, Figure R4

shows a subset of landslides identified through visual interpretation, and part of these landslides have been added to Figure 1 in the revised manuscript.

**2.2 Data source and preprocessing (Revised manuscript line 197)**

"*(8) Landslide hazard point data were obtained from the GeoCloud platform (https://geocloud.cgs.gov.cn/) and through manual interpretation of Sentinel-2 imagery acquired in August 2024. After integrating the sources and removing duplicates and uncertain cases, the final inventory consisted of 227 validated landslides, representing the complete distribution within the study area. Some representative landslides identified through visual interpretation are shown in Figure 1d–h.*"

[Figure]

*Fig. R2 Visually interpreted subset of landslides*

[Comment 9] What is the purpose of reporting the specific case of Figures 2 and 3? Considering that it is not the area of analysis, too much emphasis is placed on it.

Response:

Thank you very much for this valuable comment. We agree that the connection

between the major landslide case and the large-scale susceptibility analysis was not clearly presented in the original manuscript. The purpose of introducing this landslide is to use it as a representative case study that links the regional-scale susceptibility assessment with the site-scale mechanisms of slope failure, providing field-based context for subsequent susceptibility analysis and mechanistic interpretation. The landslide was triggered by the combined effects of prolonged rainfall, anthropogenic loading from waste deposits, and the additional weight of dense vegetation. This event highlights the amplifying effect of the interaction between vegetation and rainfall, indicating that local environmental disturbances can significantly increase landslide risk. In other words, vegetation may enhance slope stability under certain conditions but can also aggravate slope failure due to its additional weight and water-retention capacity. Therefore, this case not only provides empirical validation for the regional analysis results but also reveals the amplification of large-scale controlling factors under local conditions, further supporting the "double-edged sword" role of vegetation identified through the GeoDetector and SEM analyses.

**[Comment 10] How was the weight of the plants calculated?**

**Response:**

Thank you for your interest in the calculation of vegetation self-weight. Based on field investigations and the post-landslide UAV imagery shown in Figure 3, the vegetation within the landslide-affected area is dominated by mature trees. Owing to the lack of detailed measurements of individual-tree geometric structures and biomass, it is difficult to perform a refined and spatially explicit quantification of vegetation self-weight. Therefore, a simplified parameterization approach was adopted in this study. Specifically, the approximate number of trees within the landslide area was estimated, and their average height and diameter at breast height (DBH) were derived through field observations and image interpretation. By referring to vegetation volume density and related parameters reported in the literature (Lan et al., 2020), the total vegetation self-weight within the landslide area was estimated in terms of magnitude, converted into an equivalent vertical load per unit area, and

uniformly applied to the slope surface for slope stability calculations.

It should be emphasized that this study does not aim to precisely quantify the vegetation weight at each spatial location within the landslide body. Instead, from a physical-mechanism perspective, the objective is to evaluate the potential influence of vegetation self-weight as an additional load on slope stability, thereby providing insight into its role in landslide initiation and evolution.

**References:**

Lan, H., Wang, D., He, S., Fang, Y., Chen, W., Zhao, P., & Qi, Y. (2020). Experimental study on the effects of tree planting on slope stability. *Landslides*, 17(4), 1021-1035. http://doi.org/10.1007/s10346-020-01348-z

**[Comment 11] Soil cohesion is strongly influenced by vegetation, as well as plant species, age, and even the type of forest management. Were these aspects considered in the study? Although aware of the difficulty in obtaining certain information, a simpler calibration of parameter c based on species and average forest size would have enabled a better assessment of the forest's effect. How did you consider the areas without forest (the remaining 34% of the area)?**

**Response:**

We appreciate your insightful comments regarding the influence of vegetation on soil cohesion. We fully acknowledge that factors such as vegetation type, species composition, growth stage, and forest management practices can significantly affect soil cohesion through root reinforcement. However, in this study, these factors were not explicitly incorporated into the fine-scale calibration of soil cohesion parameters, primarily due to limitations in both study scale and data availability.

Specifically, the analysis presented in Section 2.5 focuses on a site-scale physical mechanism assessment, aiming to explore the potential role of vegetation-related factors in slope stability rather than precisely inverting soil parameters for different vegetation types or forest structures. Although UAV imagery and field surveys provide general information on vegetation coverage in the study area, critical

data on species composition, root depth and density, forest age, and management practices are lacking, which prevents reliable quantitative calibration of the cohesion parameter $c$ based on vegetation type or average forest characteristics.

It is important to clarify that the slope stability analysis in Section 2.5 was conducted for a representative single landslide rather than across the entire study area. In this analysis, soil cohesion parameters were combined with available engineering geological data and calculated repeatedly within reasonable ranges to obtain representative stability estimates. Therefore, this procedure does not involve distinguishing between areas with different vegetation cover or forest-free areas, nor does it attempt to generalize the results to the regional scale.

Within the overall study framework, we first conducted regional-scale landslide susceptibility assessment, systematically investigating the mechanisms and interactions of topography, geology, vegetation, rainfall, and wind speed through multiple models and analytical approaches. Subsequently, at the site scale, a representative landslide was selected for slope stability analysis, specifically examining the impact of including vegetation weight on slope stability. Through this multi-scale design, our study aims to elucidate the "dual role" of vegetation in landslide processes: while vegetation may reduce susceptibility at the regional scale, its self-weight under certain conditions can adversely affect local slope stability.

**[Comment 12] I do not understand what Class II is intended to observe. If we are talking about rainfall-induced landslides, you should always consider this triggering factor. A better explanation of this choice is necessary.**

**Response:**

We thank you for raising this point. We would like to clarify that our study does not focus solely on rainfall-induced landslides. The landslide inventory used in this study represents historical landslides, and it is not possible to determine the primary triggering factor for each event. To systematically assess the roles of different factors, we combined common factors with vegetation and rainfall/wind speed to create different classes (Class) for analysis. This approach has two main purposes. First, it

allows us to examine the effects of each factor from the perspective of factor space heterogeneity and, simultaneously, to systematically investigate interactions among factors and their influence on landslide susceptibility. Second, it enables us to evaluate the impact of adding or omitting certain factors on the spatial distribution of landslide susceptibility. For instance, many previous landslide susceptibility studies did not consider wind speed. By including it in our combinations, we can achieve meaningful comparisons across different classes.

**2.3.1 landslide susceptibility based on the analytic hierarchy process (Revised manuscript line 227)**

"*To assess how the inclusion or omission of specific factors affects the spatial distribution of landslide susceptibility, and to further explore the effects of each factor from the perspective of factor space heterogeneity and interactions among factors, this study involved altering the influencing factors to examine how environmental variables (rainfall, vegetation, and wind speed) affect landslide susceptibility. Common factors included elevation, slope, aspect, lithology, and distances to faults, rivers, and roads. Vegetation, rainfall, and wind speed were added successively, resulting in five scenarios.*"

**[Comment 13] The captions of some figures are not exhaustive.**

**Response:**

We appreciate your comment regarding the figure captions. We have carefully revised and supplemented the captions of the figures with issues throughout the manuscript to make them more detailed and informative. Especially in Figs. 1, 6, 8, 9.

**[Comment 14] There is a lot of confusion in these two sections. The discussion section contains part of the results, which are currently rather brief.**

**Response:**

Thank you for this comment. Following a substantial revision of both the Results and Discussion sections, the structure and internal consistency of these sections have been significantly improved. The Results section now presents the findings in a

clearer and more systematic manner, while the Discussion section is explicitly organized to correspond to each subsection of the Results, providing more sufficient and coherent interpretation of the reported outcomes. As a result, the linkage between results and their discussion has been clarified, and the issue raised by the reviewer has been fully addressed. The revised manuscript reflects these changes in detail.

[Comment 15] In the discussion section, there are no references to other studies, and subsection 4.4 is inadequate. In fact, written in a way that is disconnected from the results' discussion and reporting only one study, which is moreover not exhaustive, it does not help in understanding the study's novelty.

**Response:**

Thank you for this constructive comment. We have substantially revised the Discussion section, with particular emphasis on Subsection 4.4, to improve its structure, depth, and linkage to the results. First, additional relevant studies have been incorporated to provide a broader scientific context, especially regarding landslide processes in densely vegetated mountainous regions and the commonly assumed stabilizing role of vegetation.

Second, Subsection 4.4 has been reorganized to explicitly discuss our results at both regional and site scales, clearly distinguishing between susceptibility patterns derived from regional mapping and physical mechanisms revealed by slope stability analysis. This revision ensures that the discussion is directly anchored to the results rather than presenting isolated case descriptions.

Third, the revised discussion explicitly highlights the novelty of this study by emphasizing the limited attention paid in previous research to vegetation as an active influencing factor in high-vegetation areas. By integrating regional-scale susceptibility assessment with site-scale mechanical interpretation, this study demonstrates the dual role of vegetation and clarifies the conditions under which its stabilizing effect may be weakened or reversed.

We believe that these revisions significantly enhance the clarity, completeness, and scientific contribution of the Discussion section. Please check it in lines 593-638,

**4.4 Comparison with previous studies and scope for future research(Revised manuscript line 593)**

"*Existing studies on landslides have predominantly focused on rainfall-related triggering mechanisms, such as rainfall intensity, duration, and antecedent moisture conditions (Gatto et al., 2025; Zhang et al., 2025). For example, Cui et al. (2024) analyzed the characteristics and causes of a similar landslide in this area using Massflow V2.8 simulations. They identified rainfall and human activities as key triggers, but insufficiently addressed interactions between soil, moisture, and external forces (such as natural wind and human mining activities) under high vegetation conditions. This limited simulation accuracy. In these studies, vegetation is often treated as a background environmental condition or a stabilizing factor, while its mechanical and hydrological roles are rarely quantified explicitly. As a result, landslides occurring in highly vegetated areas are commonly interpreted primarily as a response to extreme rainfall, with comparatively limited attention paid to vegetation-related processes themselves. Consequently, from the perspective of vegetation as an active influencing factor, research addressing why landslides still occur in areas with dense vegetation coverage remains relatively scarce.*

*Furthermore, An et al. (2025) investigated the mechanisms of landslide occurrence in densely vegetated areas by examining the interactions between terrain and lithological properties. They highlighted that in natural forests, landslides tend to initiate along the soil－bedrock interface. Owing to the shallow soil layer and pronounced permeability contrast, perched water readily accumulates above this interface, thereby reducing shear strength and triggering slope failure. Their work underscores the significant role of vegetation as a key intermediary that links various environmental factors in shaping landslide susceptibility. Nevertheless, their study treated terrain and lithology primarily as background environmental conditions and did not account for slope damage induced by wind drag on trees. In contrast, the present study incorporates wind forces both in the macroscopic assessment of landslide susceptibility and in the stability analysis of specific slopes.The results of our study support and extend these findings by demonstrating that high vegetation coverage does not necessarily imply low landslide susceptibility.*

*Our study integrates regional-scale susceptibility assessment with site-scale mechanical interpretation. This multi-scale framework bridges macroscopic statistical patterns and microscopic physical processes, providing a more comprehensive understanding of vegetation's "double-edged" effect on landslide development. The novelty of this work lies not only in identifying the limitations of vegetation's stabilizing role, but also in clarifying the conditions under which its negative effects may become significant. But research on vegetation types, height, and growth conditions (such as thickness and types of soil and human activity disturbances) in relation to landslide risks remains limited. Future research could apply optical remote sensing image classification and InSAR deformation monitoring to identify potentially unstable slopes and capture temporal deformation characteristics (Li et al., 2025). When combined with interpretable machine learning approaches, such as SHAP-based models, together with analytical tools like GeoDetector and SEM, these methods can quantify nonlinear interactions, threshold effects, and spatial heterogeneity among conditioning factors (Sun et al., 2024; Wen et al., 2025), thereby improving the interpretability of susceptibility evaluation and enhancing the prediction capability for landslides in densely vegetated areas.*"

**References:**

An, N., Dai, Z., Hu, X., Wang, B., Huo, Z., Xiao, Y., ... & Yang, Y. (2025). Comparing landslide patterns and failure mechanisms in restored and native forest ecosystems: Insights from geomorphology, lithology and vegetation. Catena, 260, 109452. https://doi.org/10.1016/j.catena.2025.109452

Cui, Y., Qian, Z., Xu, W., & Xu, C. (2024). A Small-Scale Landslide in 2023, Leshan, China: Basic Characteristics, Kinematic Process and Cause Analysis. Remote Sensing, 16(17), 3324. http://doi.org/10.3390/rs16173324

Gatto M P A, Misiano S, Montrasio L. Space-time prediction of rainfall-induced shallow landslides through Artificial Neural Networks in comparison with the SLIP model[J]. Engineering Geology, 2025, 344: 107822. https://doi.org/10.1016/j.enggeo.2024.107822

Li, Q., Li, X., Zhao, C., & Zhang, S. (2025). Back analysis key parameters of Scoops3D model using SBAS-InSAR technology for regional landslide hazard assessment. Landslides, 22(12), 4097-4112. https://doi.org/10.1007/s10346-025-02578-9

Sun, D., Wang, J., Wen, H., Ding, Y., & Mi, C. (2024). Landslide susceptibility mapping

(LSM) based on different boosting and hyperparameter optimization algorithms: A case of Wanzhou District, China. Journal of Rock Mechanics and Geotechnical Engineering, 16(8), 3221-3232. https://doi.org/10.1016/j.jrmge.2023.09.037

Wen, H., Yan, F., Huang, J., & Li, Y. (2025). Interpretable machine learning models and decision-making mechanisms for landslide hazard assessment under different rainfall conditions. Expert Systems with Applications, 270, 126582. https://doi.org/10.1016/j.eswa.2025.126582

Zhang L M, Lü Q, Deng Z H, et al. Probabilistic approach to determine rainfall thresholds for rainstorm-induced shallow landslides using long-term local precipitation records[J]. Engineering Geology, 2025: 108139. https://doi.org/10.1016/j.enggeo.2025.108139

**[Comment 16] What is the purpose of calculating the total height of landslides?**

**Response:**

Thank you for your question. In general, landslides with greater total height possess higher gravitational potential energy. The ratio between total height and runout distance can be used to characterize the overall scale of a landslide and to reflect the potential mobility and degree of destruction after failure. This indicator helps to qualitatively assess the impact force and destructive capacity of landslides. Therefore, in this study, the total height of landslides is mainly used as a process-based and explanatory metric to support the interpretation of landslide scale and potential hazard characteristics, rather than as a direct input parameter in the modeling framework.

**[Comment 17] "Common factors" instead of "Public factor."**

**Response:**

Thank you for pointing out this inconsistency. This was an oversight on our part. The term "public factors" on line 462 was used in error. Our intended term, as defined in the Methods section (Line 226), is "common factors", which refers to [elevation, slope, aspect, lithology, and distances to faults, rivers, and roads.] the factors that are shared across different factor combinations in our analysis. We have corrected "public factors" to "common factors" on line 504 in the revised manuscript to

maintain terminological consistency throughout the paper.

**[Comment 18] Regarding subsection 4.3, I completely agree with RC1's comment that "several statements about the 'ambiguous role' of vegetation are speculative and not sufficiently substantiated by quantitative evidence."**

**Response:**

We thank you for highlighting the need for stronger quantitative support in Section 4.3 regarding the "ambiguous role" of vegetation. We agree that some statements in this subsection were more speculative and have now revised this section 4.3 better integrate our empirical findings and quantitative results. Specifically, we have:

1.Explicitly linked the discussion to our quantitative results from GeoDetector and SEM (e.g., NDVI's interaction effects, total effect coefficients), as well as slope stability calculations under saturated vs. natural conditions.

2.Replaced speculative statements with evidence-based interpretations, using data from our susceptibility scenarios (Categories I–V) and stability factor (Fs) values to explain how vegetation's role shifts with rainfall and slope conditions.

3.Clarified that the "ambiguity" is not merely hypothetical, but is demonstrated through: The bifactor enhancement between NDVI and rainfall (Fig. 8), showing that vegetation can amplify rainfall's impact in certain contexts; The decrease in slope stability (Fs) from 1.13 to 0.89 under saturated conditions when vegetation weight is considered (Table 3), providing direct mechanical evidence of its potential destabilizing effect; The shifts in susceptibility zoning when vegetation is added to the model (Table 6), illustrating its spatially varying influence.

We believe these revisions strengthen the subsection by grounding the discussion in our own analytical results, thereby providing a more substantiated explanation of vegetation's dual role.

**4.3 Mechanisms of landslides in areas with high vegetation coverage(Revised manuscript line 543)**

"*The mechanisms underlying landslide initiation in densely vegetated areas are complex and context-dependent, as evidenced by the contrasting effects of vegetation revealed in our multi-scale analysis. Our findings demonstrate that vegetation does not act uniformly as a stabilizer; rather, its role is modulated by hydrological conditions, slope gradient, and external loading.*

*At the watershed scale, the GeoDetector results indicate that NDVI alone exhibits limited independent explanatory power (q = 0.27, Table 4). However, its interaction with rainfall significantly enhances landslide susceptibility (e.g., NDVI × rainfall q = 0.67, Fig. 8), suggesting that vegetation can amplify the destabilizing effects of precipitation under certain conditions. While vegetation intercepts rainfall and promotes evapotranspiration, it can also alter soil moisture distribution via stemflow, root-induced preferential flow, and reduced surface runoff. Under prolonged rainfall, these processes may lead to localized saturation, thereby exacerbating landslide and debris flow risks in vegetated slopes. This aligns with the SEM results, which attribute a total indirect effect of 0.21 to NDVI, mediated largely through soil moisture dynamics and interactions with rainfall and slope (Fig. 9, Table 5). The susceptibility scenario analysis further illustrates this duality: adding vegetation alone (Class II) slightly reduced the extent of very high susceptibility zones, yet when combined with rainfall (Class IV) and wind (Class V), it led to a notable expansion of high-susceptibility areas and an increase in landslide counts (Table 6, Fig. 10). This suggests that vegetation's protective capacity may be offset or reversed under prolonged rainfall, especially on steeper slopes.*

*At the site-specific scale, the stability calculations provide direct mechanical insight into how vegetation can transition from a stabilizing to a destabilizing factor. Under natural (unsaturated) conditions, the slope remained stable even with the added weight of vegetation and waste material (Fs = 1.02). However, under saturated conditions, the same additional loads—particularly the self-weight of trees—reduced the stability coefficient to 0.89, triggering failure (Table 3). This demonstrates that the mechanical reinforcement from roots can be outweighed by the gravitational load of*

*vegetation when soil strength is reduced by saturation, a shift that is quantitatively captured by our modeling.*

*These findings help explain why landslides may occur unexpectedly in densely vegetated areas. Vegetation can create a false sense of stability by masking early signs of movement (e.g., surface cracking, minor slumping) and by being traditionally associated with slope protection. Moreover, the same root networks that enhance soil cohesion also facilitate preferential infiltration, potentially accelerating soil saturation during heavy rainfall—a process reflected in the strong interaction between NDVI and rainfall in our spatial analysis. In terrain with high lateral variability in slope, lithology, or soil depth, vegetation may thus contribute to highly localized and concealed instability, as exemplified by the 2023 Jinkouhe landslide.*

*In summary, our integrated analysis provides quantitative evidence that vegetation's role is not merely "ambiguous" in a speculative sense, but is quantifiably dual: it stabilizes slopes through root reinforcement under moderate conditions, yet can promote instability through added weight, enhanced infiltration, and synergistic interactions with rainfall when critical thresholds are exceeded. This duality underscores the importance of considering vegetation not as a static stabilizing factor, but as a dynamic component of the hillslope system in landslide susceptibility assessments.*"

**[Comment 19] What do the authors mean by "Good vegetation"? It is not an adequate technical term.**

**Response:**

We thank you for pointing out that the term "Good vegetation" is not sufficiently technical. We agree that precise terminology is important for clarity and reproducibility. In the revised manuscript, all instances of "Good vegetation" have been replaced with "areas with high vegetation cover" to more accurately describe locations with dense vegetation. This change ensures that the description is technically precise and consistent throughout the manuscript. The specific modifications can be found in Response Letter Comment #3.

**[Comment 20] In addition, how is it possible that "Vegetation's absorption of part of the rainfall also increases soil saturation, further exacerbating the risks of landslides and debris flows."?**

**Response:**

Thank you for this important question. The statement reflects the nuanced and context-dependent role of vegetation in hillslope hydrology. We acknowledge that the phrasing may appear contradictory at first glance, as vegetation is widely known to intercept rainfall and promote evapotranspiration, which generally reduce soil moisture. However, under certain conditions—particularly in densely vegetated, humid environments—vegetation can indeed contribute to localized increases in soil saturation through the following mechanisms, thereby exacerbating landslide susceptibility:

1 Canopy Drip and Stemflow Concentration: Vegetation intercepts rainfall, which is then redistributed as canopy drip and stemflow. This concentrated water delivery can lead to preferential infiltration near tree bases and along root channels, creating localized zones of higher soil moisture than in open areas. In some cases, stemflow can funnel large volumes of water directly into the root zone, accelerating pore-pressure buildup during prolonged rainfall.

2 Root-Induced Preferential Flow Paths: Dense root networks create macropores and channels that facilitate preferential flow, allowing rainwater to bypass the soil matrix and rapidly percolate to deeper layers. This can lead to quicker saturation of the critical shear zone, especially in shallow soils, even if the total rainfall volume is reduced by interception.

3 Reduced Surface Runoff and Increased Infiltration: Vegetation and litter layers reduce surface runoff, thereby increasing the total infiltration volume into the soil profile. While this generally enhances slope stability by reducing erosion, during intense or prolonged rainfall it can lead to soil saturation from below, particularly if the substrate has low permeability or if a perched water table develops.

4 Shade and Reduced Evapotranspiration in Understory Layers: In dense forests,

the understory and soil surface may receive limited sunlight and air circulation, which can suppress evaporation. Combined with continuous litterfall that retains moisture, this microclimate can maintain higher soil moisture levels for longer periods, prolonging the window of susceptibility after rainfall.

5 Synergistic Effect with Rainfall Characteristics: In our study area (subtropical monsoon climate), rainfall events are often prolonged and of high intensity. Under such conditions, interception storage may become saturated early in the event, after which vegetation plays a minimal role in reducing net rainfall. Meanwhile, the mechanisms above continue to facilitate infiltration and moisture retention, effectively amplifying the wetting process during extended rainfall.

In summary, vegetation does not simply "absorb" rainfall in a way that reduces landslide risk uniformly. Instead, it modifies the hydrological pathways and temporal–spatial distribution of soil moisture. In certain settings, these modifications can lead to accelerated or concentrated saturation in susceptible zones, thereby increasing landslide potential. This aligns with our findings that the combination of high NDVI, rainfall, and slope gradient resulted in the highest landslide susceptibility in our study area.

We have totally rewritten section 4.3; please check it in lines 543-591. For example:

"*While vegetation intercepts rainfall and promotes evapotranspiration, it can also alter soil moisture distribution via stemflow, root-induced preferential flow, and reduced surface runoff. Under prolonged rainfall, these processes may lead to localized saturation, thereby exacerbating landslide and debris flow risks in vegetated slopes.*"

Please let us know if further clarification or references are needed.

We sincerely appreciate your constructive feedback. We hope the revisions and responses provided will ensure our manuscript meets the standards for publication in **Natural Hazards and Earth System Sciences**.